# On provable privacy vulnerabilities of graph representations

Ruofan Wu[*§], Guanhua Fang[*‡],
Mingyang Zhang[§], Qiying Pan[¶], Tengfei Liu[§†], and Weiqiang Wang[§]

[§]Ant Group
[‡]Fudan University
[¶]Shanghai Jiao Tong University
{ruofan.wrf, zhangmingyang.zmy, aaron.ltf, weiqiang.wwq}@antgroup.com
fanggh@fudan.edu.cn, sim10_arity@sjtu.edu.cn

## Abstract

Graph representation learning (GRL) is critical for extracting insights from complex network structures, but also raises security concerns due to potential *privacy* vulnerabilities in these representations. This paper investigates the structural vulnerabilities in graph neural models where sensitive topological information can be inferred through edge reconstruction attacks. Our research primarily addresses the theoretical underpinnings of similarity-based edge reconstruction attacks (SERA), furnishing a non-asymptotic analysis of their reconstruction capacities. Moreover, we present empirical corroboration indicating that such attacks can (almost) perfectly reconstruct sparse graphs as graph size increases. Conversely, we establish that sparsity is a critical factor for SERA's effectiveness, as demonstrated through analysis and experiments on (dense) stochastic block models. Finally, we explore the resilience of private graph representations produced via noisy aggregation (NAG) mechanism against SERA. Through theoretical analysis and empirical assessments, we affirm the mitigation of SERA using NAG. In parallel, we also empirically delineate instances wherein SERA demonstrates both efficacy and deficiency in its capacity to function as an instrument for elucidating the trade-off between privacy and utility. [1]

## 1 Introduction

With the surging developments of graph representation learning (GRL) [15], there has been growing apprehensions concerning the security challenges associated with the deployment of graph neural models in real-world scenarios [10]. GRL models harness the topological information of the underlying graph for producing high-quality predictions or graph representations. Meanwhile, these models bear the risk of inadvertently divulging the same topological information through the graph representations they produce. Such kind of security risks have been empirically validated through the examination of the attacking performance of edge reconstruction algorithms [11, 17, 34, 46], among which a simple form of attack based solely on the representation similarity of node pairs is shown to achieve strikingly strong performance, without the requirement of additional knowledge like encoder architecture or auxiliary datasets [17].

---

[*]Equal contribution

[†]Corresponding author: aaron.ltf@antgroup.com

[1]Code available at https://github.com/Rorschach1989/gnn_privacy_attack

38th Conference on Neural Information Processing Systems (NeurIPS 2024).

Despite the empirical evidence of topological vulnerabilities of graph representations, theoretical explanations delineating the effectiveness of such attacks remain largely unexplored: As demonstrated in previous studies [11, 17], similarity-based attacks are remarkably effective against *sparse* graphs that exhibit a generalized homophily pattern, i.e., there exists a significant correlation between the similarity of node features and edge adjacency information. This phenomenon posits that *feature similarity* may serve as a confounding factor, potentially impacting the efficacy of similarity-based attacks. It is therefore valuable to understand the influence of graph properties, such as feature similarity and sparsity, on the edge reconstruction process of the attacking procedures.

Beyond their capability in characterizing the vulnerabilities of representations, attacking algorithms may also function as empirical attestations of privacy-preserving inference protocols that fulfill formal privacy guarantees such as differential privacy [9, Section 4]. As an illustrative case, membership inference attacks can be employed for auditing differential privacy [31]. Since edge reconstruction is equivalent to edge membership inference on graphs [43], it is thus pertinent to explore the performance of similarity-based attacks when confronted with privacy-preserving graph representations [30, 36].

In this paper, we take initial steps toward a principled understanding of structural vulnerabilities of graph representations under the **s**imilarity-based **e**dge **r**econstruction **a**ttack (hereafter abbreviated as SERA) which forms a realistic threat in many practical scenarios such as vertical federated learning [36]. In particular, we establish the following theoretical as well as empirical findings:

(i) **Success modes of SERA** Through applying SERA to sparse random graphs equipped with independent random node features, we show that SERA provably reconstructs the input graph via a non-asymptotic analysis. The result indicates that feature similarity is not necessary for SERA to succeed. We conduct both synthetic experiments as well as real-world data evaluations to empirically validate our theory.

(ii) **Failure modes of SERA** We show, through theoretical analysis and corroborative synthetic experiments, performance lower bounds when applying SERA to stochastic block models (SBM) with independent random node features: When the underlying SBM has $\Theta(1)$ intra-group connection probability, edge recovery through graph representations becomes provably hard.

(iii) **Mitigation of SERA** We assess the resilience of SERA using noisy aggregation (NAG) as the privacy protection mechanism. Theoretical guarantees of NAG are established which further extends previous results, accompanied by extensive empirical evaluations to corroborate our theoretical assertions. Intriguingly, our findings reveal instances wherein NAG provides significant resistance to SERA, even under some scenarios where it only guarantees very weak privacy. Such discoveries delineate the circumstances that elucidate both the strengths and limitations of SERA as a privacy auditing tool.

## 2 Related works

Typically, there exist two categories of private information that may potentially be compromised during the training or deployment phases of graph neural network models: The (sensitive) node attributes and the adjacency relation between nodes. In this paper we focus on the later category since edge adjacency relations are less informative, i.e., for each pair of nodes, the existence of an edge constitutes only a single bit of information.

### 2.1 Edge reconstruction attacks on graph-structured data

Contemporary developments on edge reconstruction attacks differ significantly in their conceptualization of adversaries, particularly in terms of their capabilities [45, 44] and the extent of prior knowledge they possess about the GRL model and the underlying graph dataset [17]. The mechanism of SERA was first proposed in [11] and later studies in [17]. Empirical evidences suggest that with only black-box access to node representations, the SERA mechanism obtains a high success rate (AUC $> 0.9$ for the Citeseer dataset). Subsequent developments have explored stronger attacks under more powrful adversaries. In [17] the authors investigated the impact of an adversary's prior knowledge, including the possession of node features, partial graph structure, and access to a shadow dataset, on the success rate of corresponding attack strategies. Inspire by information bottleneck,[46] improves SERA via carefully exploiting intermediate representations produced by GNNs. Notably,

despite the adversaries in [17, 46] being equipped with substantially more information compared to SERA, the resulting enhancement in attack performance exhibited by these adversaries demonstrates only marginal improvements relative to SERA. The GraphMI attack [45] disables the adversary from being able to acquire node representations but instead requires access to node features and labels, as well as white-box access to the GNN model. Recent works explored influence-based attacking schemes, wherein the adversary is allowed to alter the graph information: The LinkTeller attack [34] manipulates node features while [23] infiltrates the underlying graph with malicious nodes.

## 2.2 Theoretical explorations in graph recovery from neural representations

In [6], the authors proposed an algorithm that provably recovers graph structure based on representations generated via DeepWalk, which is a factorizaton-based procedure and different from GNN-produced representations. In [43] the authors showed that when block structure exists in the underlying graph, the performance of SERA is uneven across node in different blocks. In [46], the authors use information-theoretic arguments to construct more powerful attacks than SERA. Nevertheless, the aforementioned studies did not provide a theoretical rationale for the practical vulnerabilities manifested as a result of the SERA. In a contemporary work [8], the authors derived generalization bounds of linear GNN under the link prediction context assuming the underlying graph generated by a moderately sparse graphon model.

## 2.3 Privacy protection against edge reconstruction attacks

Edge differential privacy (EDP) [26] is the most popular privacy notion that offers a formal protection against edge reconstruction attacks. Standard private training algorithms like DPSGD [1] may produce GNN models that is provably private in the sense that membership information of any individual training sample is limitly disclosed. [2] However, such approaches do not provide privacy during *inference* time [7]. Protection mechanisms against inference-time adversaries are mostly based on noisy version of GNN encoding such as edge-wise randomized response [34] that provides very strong privacy protection yet being overly destructive to model utility. Noisy aggregation (NAG) mechansims [30, 36, 7] are recently proposed that empirically achieves better privacy-utility trade-offs. Inspired by the information bottleneck principle, [33, 46] proposed to use regularization or saddle-point optimization techniques to control privacy leakage. Yet these proposals are not principled in theory.

# 3 Preliminaries

**Setup and notations** Consider an undirected graph $G = (V, E)$ with $n = |V|$ nodes associated with node features $X \in \mathbb{R}^{n \times d}$. Denote $A$ as the corresponding adjacency matrix and $D$ as the diagonal matrix with the $v$-th diagonal entry being the degree of node $v$. In this paper, we will study *victim models* taking forms of graph neural encoders. Our vulnerability analysis predominantly centers on the *linear graph neural network* [35] architecture which has been widely adopted in previous theoretical studies on graph neural networks [3, 39, 37, 8]. Specifically, the node representation matrix of an $L$-layer linear GNN is computed as:

$$H^{(L)} = \left( (D + I)^{-1} (A + I) \right)^L XW, \tag{1}$$

where the identity matrix is added for ensuring self-loops, and $W \in \mathbb{R}^{d \times d}$ is the weight matrix. Throughout this paper, we will assume the node feature dimension and the hidden dimension to be equal to $d$ and refer to this as the feature dimension, as otherwise we may add an extra input projection to fulfill this requisite. We further denote $\|W\|_{\text{op}}$ and $\kappa(W)$ as the operator norm (i.e., largest singular value) and condition number (i.e., the ratio of largest and smallest singular value) of matrix $W$.

**Threat model** We assume the adversary knows the node set $V$ and is able to inquire node representations of an arbitrary node subset $V_{\text{victim}} \subset V$. Hereafter we will refer to the subgraph induced via $V_{\text{victim}}$ as the *victim subgraph* $G_{\text{victim}} = (V_{\text{victim}}, E_{\text{victim}})$. The goal of the adversary is to recover an arbitrary fraction of $E_{\text{victim}}$ based on the acquired node representations

---

[2]Note that this require a careful sensitivity analysis with respect to the correct privacy model like EDP.

$H_{\text{victim}}^{(L)} = \{h_v^{(L)}, v \in V_{\text{victim}}\}, L > 0$. We identify two representative scenarios that underscore the potential threat by such adversaries: The first scenario is API-style deployments of graph representations [34], wherein an adversary might query the node representations for a set of nodes using their node identifiers, with this particular subset of nodes constituting the victim nodes. The second scenario pertains to a two-party vertical federated learning (VFL) context [36], wherein the graph topology retained by party A is deemed confidential. Under such a setup, the privacy threat materializes as party B might adhere to the VFL protocol while simultaneously being curious about the topology. Note that the capabilities of the adversaries posited herein are intentionally constrained by denying them access to both the raw node features $X$ and the model parameters. Additionally, the objectives of the adversary are decidedly ambitious, aiming at the potential recovery of the entire suite of edges within the victim subgraph. A more in-depth discussion regarding the threat model and the potent capabilities of the adversary is deferred to appendix B.1.

The SERA is based on a similarity measure sim, with the adjacence relation between node $u$ and node $v$ inferred as

$$\widehat{A}_{uv}^{\text{SERA}}(\tau) = \mathbf{1}\left(\text{sim}\left(h_u^{(L)}, h_v^{(L)}\right) \geq \tau\right), \tag{2}$$

where we denote $\mathbf{1}(\cdot)$ as the indicator function. In this paper we will be primarily interested in two similairty measures: The cosine similarity $\text{cos}(x, y) = \langle x, y \rangle / (\|x\|_2 \|y\|_2)$ and correlation similarity $\text{corr}(x, y) = \langle x - \bar{x}, y - \bar{y} \rangle / (\|x - \bar{x}\|_2 \|y - \bar{y}\|_2)$, which is essentially a centered version of cosine similarity ($\bar{x}, \bar{y}$ are coordinate-wise averages of $x$ and $y$) defined for node representations with dimension greater than 1. The cutoff threshold $\tau$ is allowed to depend on the embedding set $H_{\text{victim}}$ but is uniform across all edge decisions. Hereafter without misunderstandings, we will drop the superscript and denote $\widehat{A}(\tau)$ as the reconstructed adjacency matrix under threshold $\tau$. To measure the performance of the attack, we use false positive rate (FPR) and false negative rate (FNR) defined as

$$\text{FPR}_{\widehat{A}}(\tau) = \frac{\sum_{u,v} \mathbf{1}\left(\widehat{A}_{uv}(\tau) = 1\right) \mathbf{1}\left(A_{uv} = 0\right)}{\sum_{u,v} \mathbf{1}\left(A_{uv} = 0\right)}, \text{FNR}_{\widehat{A}}(\tau) = \frac{\sum_{u,v} \mathbf{1}\left(\widehat{A}_{uv}(\tau) = 0\right) \mathbf{1}\left(A_{uv} = 1\right)}{\sum_{u,v} \mathbf{1}\left(A_{uv} = 1\right)}. \tag{3}$$

We further define the error rate ERR as the summation of FPR and FNR. Employing these metrics facilitates a more nuanced characterization of attack performance, particularly when the underlying graph is sparse. An alternate metric that is often used in practice [17] is the area under the receiver operating characteristic curve (AUROC) metric

$$\text{AUROC}_{\widehat{A}} = \int_0^1 \left(1 - \text{FNR}_{\widehat{A}}\left(\text{FPR}_{\widehat{A}}^{-1}(s)\right)\right) ds \tag{4}$$

which quantifies the aggregate performance of $\widehat{A}$ by integrating the trade-off between the false positive rate and the false negative rate across different thresholds.

Intuitively, the success of SERA is determined by the correlation between node representation similarity and edge presence. Previous empirical observations demonstrate the effectiveness of SERA against graphs that exhibit strong correlations between node feature similarity and edge presence [17]. We will refer to such kinds of graphs as being homophilous in a generalized sense [18, 22]. We defer a more formal introduction to homophily measrues to appendix B.2. Due to the message-passing nature of GNN encoders, it is intuitively reasonable that recursive aggregation of node representations strengthens the correlation and results in successful edge reconstructions. However, it is non-trivial whether SERA mechanism may succeed in the absence of the aforementioned generalized homophily pattern, which motivates our first analysis.

## 4 SERA against sparse random graphs

In this section, we study the behavior of SERA with the underlying (victim) graph generated according to a *sparse random graph*. Here, the adjacency matrix is generated such that each entry is independently distributed (up to symmetric constraints $A_{uv} = A_{vu}$) following a Bernoulli distribution $A_{uv} \sim \text{Ber}(p_{uv})$. We focus on the sparse regime and allow $p_{uv}$ to depend on $X_u$ and $X_v$. We further assume that the node features $X_v$'s are generated i.i.d. according to an isotropic Gaussian distribution $X_v \sim N(0, I_d)$. It follows that the correlation of node feature similarity and edge presence is zero. The following theorem characterizes the effectiveness of SERA under the sparse random graph setup.

**Theorem 4.1.** *Let $C_1, C_2$ be a universal constants. Assume the following:*

(i) *The graph generation mechanism satisfies $\sum_{u \in V} p_{uv} < C_1 \log n$ holds for all $v \in V$.*

(ii) *The depth of GNN encoder $L$ and the feature dimension $d$ satisfies $d \gg (C_2 \log n)^{6L+2} \log n$.*

(iii) *The condition number of the GNN encoder weight satisfies*

$$(\kappa(W))^2 \leq \frac{1}{8(C_2 \log n)^{3L}} \sqrt{\frac{d}{\log n}}. \tag{5}$$

*Then there exists a threshold $\tau = \Theta\left(\frac{1}{(C_2 \log n)^{2L}}\right)$ such that with probability at least $1 - \frac{2}{n^2}$, the following holds for* **SERA** *with the similarity measure chosen either as* **cos** *or* **corr***:*

$$\textsf{FNR}_{\widehat{A}}(\tau) = 0, \ \textsf{FPR}_{\widehat{A}}(\tau) \leq \frac{(C_2 \log n)^{2L}}{n}. \tag{6}$$

*Consequently, on the above set of events we have $\textsf{AUROC}_{\widehat{A}} \geq 1 - \frac{(C_2 \log n)^{2L}}{n}$.*

Theorem 4.1 implies that, even when **SERA** can not borrow strength from the homophily nature of the underlying graph, it is able to produce accurate reconstructions when the graph is sufficiently *large and sparse*, with the sparsity defined in the sense that each node has at most $O(\log n)$ neighbors on average. An additional intriguing implication from theorem 4.1 pertains to the dependence of reconstruction performance on the GNN encoder depth $L$: Provided that the node feature dimension is sufficiently large, the reconstruction performance degrades when the depth of the encoder increases, which is related to the renowned phenomenon of oversmoothing in GNN literature [37]. Intuitively, as the depth of GNN encoders increases, the resulting node representations tend to converge [27], becoming less distinct from one another. This convergence diminishes the discriminative capacity of similarity metrics, thereby affecting the attack performance.

*Remark* 4.2 (Practicality). Theorem 4.1 requires the node feature dimension $d$ to grow in a polylog($n$) rate, a condition which may not consistently align with practical scenarios. At present, this requirement is a byproduct of our proof strategy. In section 7.1 we will further examine the implications of feature dimensionality. The existence of a threshold that theorem 4.1 manifests might not guide the choice of threshold in practice. Instead, we may rely on heuristics or side-information [17] to determine the threshold. Furthermore, Theorem 4.1 posits that the efficacy of **SERA** is contingent upon a reasonable conditioned weight matrix $W$. We will empirical validate this claim in section 7, wherein we demonstrate robust reconstruction capabilities of the **SERA** across diverse scenarios including when the weight matrix $W$ is a fixed entity, when it is subject to random initialization, or when it has undergone extensive training iterations utilizing datasets from real-world supervised learning contexts.

## 5 SERA against dense SBMs

In this section, we reveal the limitation of **SERA** by constructing a reconstruction problem that is provably hard. We consider the following stochastic block model (SBM) [2], where each node is assigned a community membership from one of $K$ groups $k(v) \in [K]$. The $(u, v)$-th entry of the adjacency matrix is generated as

$$A_{uv} \sim \begin{cases} \text{Ber}(p), & \text{if } k(u) = k(v) \\ \text{Ber}(q), & \text{otherwise} \end{cases}. \tag{7}$$

For ease of presentation, we further assume that the groups share the same size, i.e., $n$ is a multiple of $K$. Denote the generation mechanism as $G \sim \mathcal{G}_{\text{sbm}}(n, K, p, q)$. We have the following result:

**Theorem 5.1.** *Let $G \sim \mathcal{G}_{sbm}(n, K, p, q)$ and $p = \Theta(1)$. Assume the GNN encoder to be of depth $L$ and feature dimension $d \gg \max\{\log n/p^2, K^2 \log^3 n\}$ with the weight matrix being the identity matrix. Then with probability at least $1 - 1/n^2$, for any fixed $\tau \in [0, 1]$, one of the following three statements must hold for* **SERA** *with similarity measure chosen either as* **cos** *or* **corr***:*

*(i)* $\mathsf{FPR}_{\widehat{A}}(\tau) \geq \frac{1-p}{2K}$ *and* $\mathsf{FNR}_{\widehat{A}}(\tau) \geq \frac{q}{2}$.

*(ii)* $\mathsf{FPR}_{\widehat{A}}(\tau) \geq \frac{1-p}{2K} + \frac{1-q}{2}$.

*(iii)* $\mathsf{FNR}_{\widehat{A}}(\tau) \geq \frac{p}{2K} + \frac{q}{2}$.

According to theorem 5.1, given any cutoff threshold if the within-group connection probability is of the order $\Theta(1)$ and the number of groups $K$ does not diverge (Otherwise, we will return to the sparse regime in section 4) , the performance of SERA measured by error rate ERR is lower bounded by non-vanishing constants when the feature dimension is sufficiently large. The theorem characterizes the inherent limitations of SERA when the underlying graph is dense. As $K$ gets large, the lower bound of false positive/negative rate decreases. It indicates that SERA is more successful when the graph is less connected.

*Remark* 5.2. Alternatively, we may interpret theorem 5.1 as unveiling instances where SERA is constrained to revealing only population-level relational information—such as the affiliation of two nodes to a common group—rather than identifying the existence of specific edges when the underlying graph is dense and admits certain group structures.

## 6 Defense by noisy aggregation: From theory to practice

Having demonstrated the susceptibility of GNN representations to SERA, it becomes an intriguing research question to examine the behavior of SERA within the context of privacy-preserving GRL: In this section, we explore the defensive efficacy of noisy aggregation (NAG), which has been proposed recently as a provably privacy-preserving algorithm [30, 36] under the edge differential privacy model [26]. Concretely, we study an $L$-layer noisy GNN with the $l$-th layer computed recursively as:

$$H_v^{(l)} = \mathsf{Act}\left(\mathsf{AGG}\left(W_l H_u^{(l-1)} / \left\|H_u^{(l-1)}\right\|_2, u \in \overline{N(v)}\right) + \epsilon\right), \epsilon \sim N(0, \sigma^2 I_d), \tag{8}$$

where $\overline{N(v)} := N(v) \cup \{v\}$ denotes node $v$'s extended neighborhood and $H_v^{(l)}$ denotes the representation of node $v$ at the $l$-th layer. The aggregation mechanism AGG is a permutation invariant function that defines the message-passing process and Act is some (possibly) non-linear transform. Intuitively, the NAG methodology can be understood as a privatization protocol that incorporates both a normalization step and an additive Gaussian perturbation phase into the conventional message-passing framework, which typically forms the backbone of a GNN. In this paper, we consider 5 representative GNN architectures that allows NAG privatization: GCN [20], GAT [32], SAGE [14] with mean or max pooling, and GIN [38] with their formal definition deferred to appendix B.3. The following theorem characterizes the defensive capability of NAG:

**Theorem 6.1.** *For any graph $G$ and SERA under any type of similarity measures, the inference error regarding any specific edge is lower bounded by:*

$$\min_{u \in V, v \in V} \left[\mathbb{P}\left(\widehat{A}_{uv} = 1 | A_{uv} = 0\right) + \mathbb{P}\left(\widehat{A}_{uv} = 0 | A_{uv} = 1\right)\right] \geq 1 - \sqrt{1 - \exp\left(-C \frac{\sum_{l \in [L]} \|W_l\|_{op}^2}{\sigma^2}\right)}. \tag{9}$$

*Here the constant $C$ depends on the AGG mechanism of the GNN. In particular, for some standard GNN architectures we have: $C_{GCN} = C_{MEAN\text{-}SAGE} = C_{GIN} = 1$ and $C_{GAT} = C_{MAX\text{-}SAGE} = 4$.*

Theorem 6.1 augments existing literature in the sense that it extrapolates upon prior analyses [30, 36] by generalizing to a broader range of aggregation mechanisms, thereby encompassing the vast majority of foundational components integral to modern GNN models. Theorem 6.1 indicates that for *any* node pairs in any graph, the summation of type-I error and type-II error (in the language of binary hypothesis testing [21]) incurred by *any* SERA adversary is lower bounded by a constant, which will be significantly above zero when the noise scale is of the same order to the operator norms of the weight matrices of the GNN encoder. In fact, theorem 6.1 holds against a much stronger family of adversaries, which we discuss in appendix C.3.

**Empirical proctection of NAG** implementing NAG with a large noise scale according to theorem 6.1 may seriously degrade model efficacy. Contemporary insights [5] suggest that strict adherence to theoretical prescripts may not always be necessary, especially in the face of empirical adversaries whose capabilities may not rise to the level presumed by the defense mechanisms postulated. In this

paper we conduct a careful empirical investigation to assess the privacy-utility trade-off of NAG, with privacy evaluated by the SERA adversary. This investigation could also provide empirical evidence of the SERA's viability as a tool for auditing private GRL algorithms [9]. Furthermore, theorem 6.1 identifies a key determinant of the theoretical privacy bound for NAG —the relative scale of the weight norms regarding the noise intensity. In light of this observation, we propose two distinct noise-infused training paradigms:

**Unconstrained scheme** We choose a fixed noise scale $\sigma$ during both training and inference no constraints over the weights. The resulting model might not produce meaningful privacy guarantees in the sense of theorem 6.1 as the operator norms of weights are determined by the training dynamics.

**Constrained scheme** We choose a fixed noise scale $\sigma$ during both training and inference and use normalization techniques [25] to provide a priori control of model weights, thereby providing tighter control of formal privacy level according to theorem 6.1.

We will empirically inspect the protection of NAG representations trained via both unconstrained and constrained schemes against SERA in section 7.3.

*Remark* 6.2 (Alternative defenses). Beyond the scope of NAG, alternative defense mechanisms offer demonstrable protection assurances, one notable example being edge-wise randomized response (EdgeRR). A comparison with such alternatives is reported in appendix D.5. Preliminary experimental comparisons indicate that NAG customarily realizes a more favorable balance between privacy and utility.

*Remark* 6.3 (Impact of depth $L$). Theorem 6.1 posits that the privacy guarantees furnished by NAG diminishes with an increment in model depth, which is underpinned by the composition theorems of privacy analysis [12, 24]. An extensive discussion concerning the implications of GNN architectural design on the privacy-utility trade-off, particularly as it pertains to the depth of GNN models trained with NAG, will be provided in appendix E.

## 7 Experiments

In this section, comprehensive empirical studies are conducted to evaluate the effectiveness of SERA against both non-private and private node representations. By default, cos is employed as the standard measure of similarity across all experiments. The results corresponding to the use of corr as a metric were found to align with those obtained from cos, a concurrence that aligns with observations from [17]. This investigation is oriented around three core research questions:

**RQ1 (Efficacy of SERA on Sparse Graphs):** We evaluate SERA on synthetic datasets generated according to theorem 4.1, in addition to $8$ real-world datasets to substantiate the effectiveness of SERA.

**RQ2 (Deficiency of SERA on Dense Graphs):** We evaluate SERA on synthetic stochastic block models to corroborate the theoretical assertions in theorem 5.1.

**RQ3 (Mitigation of SERA through NAG):** We evaluate SERA on privacy-enhanced node representations across three benchmark datasets generated using NAG with varied levels of noise. The outcomes affirm NAG's capacity for privacy preservation while concurrently delineating the limitations of SERA as a tool for privacy auditing.

**Evaluation Metrics:** Predominantly, this section documents the performance of attacks using the AUROC metric. A more expansive presentation of the results, inclusive of both AUROC and ERR, is postponed to appendix D.

### 7.1 Efficacy of SERA on sparse graphs

**Erdős–Rényi experiments** In our first experiment, we test SERA on graph representations produced by (1) over Erdős–Rényi graphs with edge probability $p = \frac{\log n}{n}$ with graph size $n \in \{100, 500, 1000\}$, which is a representative random graph model with controllable sparsity level. We set the weight to be the identity matrix and further present results under random weights in appendix D.1. We vary the feature dimension $d \in \{2^j, 2 \leq j \leq 11\}$ and network depth $1 \leq L \leq 10$ in order to obtain a fine-grained assessment of SERA. We present the evaluations in figure 4. The results corroborate with our theoretical developments: We demonstrate that SERA is able to achieve near-perfect reconstruction of all edges *only* in the "large $d$, small $L$" regime. Notably, we find

Table 1: Performances of SERA on eight datasets measured by AUROC metric (%). The feature homophily $\mathcal{H}_{\text{feature}}(G, X) = \frac{1}{|E|} \sum_{(u,v) \in E} \cos(X_u, X_v)$ is an alternate measure of correlation between feature similarity and edge presence. For each setup, the results (in the form of mean$_{\pm\text{std}}$) are obtained via 5 random trials.

| | Squirrel | Chameleon | Actor | Cora | Citeseer | Pubmed | Products | Reddit |
|---|---|---|---|---|---|---|---|---|
| $\mathcal{H}_{\text{feature}}$ | 0.01 | 0.01 | 0.16 | 0.17 | 0.19 | 0.27 | 0.01 | 0.12 |
| $\widehat{A}^{\text{FS}}$ | 46.2 | 55.2 | 44.7 | 80.3 | 87.4 | 87.6 | 52.0 | 95.9 |
| Victim model | $\widehat{A}^{\text{SERA}}$, non-trained | | | | | | | |
| LIN($L=2$) | $72.8_{\pm0.0}$ | $76.1_{\pm0.2}$ | $73.1_{\pm0.1}$ | $93.1_{\pm0.4}$ | $92.5_{\pm0.9}$ | $93.9_{\pm1.2}$ | $97.2_{\pm0.3}$ | $96.4_{\pm0.1}$ |
| LIN($L=5$) | $72.6_{\pm0.0}$ | $76.0_{\pm0.3}$ | $73.0_{\pm0.2}$ | $95.9_{\pm0.6}$ | $93.8_{\pm0.4}$ | $96.0_{\pm0.6}$ | $99.2_{\pm0.1}$ | $95.4_{\pm0.1}$ |
| GCN($L=2$) | $87.3_{\pm0.3}$ | $87.9_{\pm0.4}$ | $87.1_{\pm0.6}$ | $99.8_{\pm0.1}$ | $99.9_{\pm0.0}$ | $99.7_{\pm0.0}$ | $99.6_{\pm0.0}$ | $97.3_{\pm0.1}$ |
| GCN($L=5$) | $82.1_{\pm0.3}$ | $84.3_{\pm0.9}$ | $84.1_{\pm0.9}$ | $99.4_{\pm0.2}$ | $99.9_{\pm0.0}$ | $99.5_{\pm0.1}$ | $99.2_{\pm0.1}$ | $96.1_{\pm0.2}$ |
| Victim model | $\widehat{A}^{\text{SERA}}$, trained | | | | | | | |
| LIN($L=2$) | $74.6_{\pm0.0}$ | $75.0_{\pm0.3}$ | $59.9_{\pm0.7}$ | $94.6_{\pm0.1}$ | $93.7_{\pm0.1}$ | $89.0_{\pm0.1}$ | $91.6_{\pm0.2}$ | $94.7_{\pm0.1}$ |
| LIN($L=5$) | $74.1_{\pm0.3}$ | $76.9_{\pm0.2}$ | $61.6_{\pm0.7}$ | $94.8_{\pm0.3}$ | $93.3_{\pm0.3}$ | $88.4_{\pm0.9}$ | $98.6_{\pm0.1}$ | $92.3_{\pm0.2}$ |
| GCN($L=2$) | $79.4_{\pm0.4}$ | $82.3_{\pm0.3}$ | $78.5_{\pm0.8}$ | $97.8_{\pm0.1}$ | $99.0_{\pm0.0}$ | $89.2_{\pm0.3}$ | $94.5_{\pm0.1}$ | $95.1_{\pm0.1}$ |
| GCN($L=5$) | $77.4_{\pm0.6}$ | $80.6_{\pm0.8}$ | $78.4_{\pm0.6}$ | $97.4_{\pm0.3}$ | $98.7_{\pm0.2}$ | $92.6_{\pm0.4}$ | $98.4_{\pm0.1}$ | $95.0_{\pm0.1}$ |

SERA to be less successful under relatively deep network architectures (i.e., $L \geq 5$) when the feature dimension is sufficiently large. Yet the behaviors in small $d$ regimes appear to be less predictable. [3] Furthermore, the influence of the feature dimension appears to be more pronounced than that of the network depth. This suggests that a greater number of features, despite their independence from graph topology, lead to potentially more privacy risks as transmitted through GNN representations. Conversely, augmenting the network depth does not necessarily correlate with an elevation in the success rate of SERA.

**Real-world data experiments** Given that the Erdős–Rényi model may not sufficiently capture the complexity of real-world graph structures, we evaluated the SERA algorithm on 8 diverse real-world graph datasets that exhibit contrasting patterns of feature similarity and edge formation. The analysis comprises the well-known Planetoid datasets [41], which are distinguished by their high homophily; the heterophilic datasets Squirrel, Chameleon, and Actor [29], which demonstrate a weak feature-edge correlation; and two larger-scale datasets, namely Amazon-Products [42] and Reddit [14]. Dataset statistics are comprehensively detailed in appendix B.2. Half of the datasets analyzed manifest a strong positive correlation of feature similarity and edge presence, which is measured via the AUROC of the estimator $\widehat{A}^{\text{FS}}_{uv}(\tau) = \mathbf{1}(\cos(X_u, X_v) \geq \tau)$, while the other half show negligible correlations, an observation underscored in the baseline ($\widehat{A}^{\text{FS}}$) row of table 1. In all evaluations, we standardize the hidden dimension to $d = 128$, with the number of GNN layers adjusted to $L \in 2, 5$. Our analysis extends beyond the linear aggregation scheme (1) to encompass four additional prominent GNN architectures: GCN [20], GAT [32], GIN [38], and SAGE [14]. To discern the effect of training dynamics on the potency of attacks, we delineate results for both pre-training (i.e., random initialization) and post-training stages. A precise account of training methodologies can be found in appendix D.2. Results pertaining to the linear GNN (LIN) and GCN are presented in table 1, with a comprehensive evaluation reserved for appendix D.2. We have the following observations:

**Homophily is not necessary for SERA to succeed:** The efficacy of SERA on the Planetoid datasets aligns with expectations. However, the outcomes from 4 heterophilic datasets illuminate significant privacy risks, despite a vacuous association between feature resemblance and edge formation. Notably, the Squirrel and Actor datasets, which demonstrate a mild negative feature-edge correlation, are still subject to substantial privacy breach, particularly with nonlinear models. These empirical findings support our theoretical assertion that a graph's sparsity plays a more pivotal role in its susceptibility to edge reconstruction attacks than the degree of homophily it exhibits. Moreover, in instances of

---

[3] In our analysis, one primary mathematical tool is the concentration of inner products of two Gaussian vectors, which is highly dependent on the dimension of the two vectors (i.e., the feature dimension). When the concentration is insufficient (a consequence of small $d$), our analysis would then be no longer correct and this partly explains why the attacking performance is limited in small $d$ regimes.

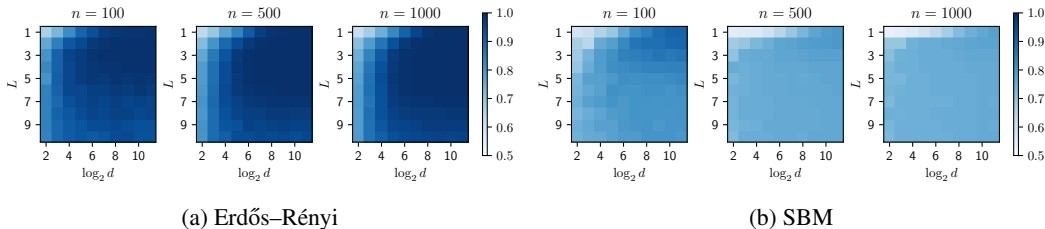

| (a) Erdős–Rényi | (b) SBM |

Figure 1: Attacking efficacy of SERA over sparse Erdős–Rényi graphs and dense SBM graphs, with performance measured in AUROC metric averaged over 5 random trials for each configuration.

comparatively denser networks, such as the Reddit dataset, the homophily of features can be exploited to mount more sophisticated attacks.

**Efficacy of Linear GNNs as Proxies for Nonlinear Counterparts:** Evidence presented in table 1 suggests that the trends exhibited by linear GNN models are broadly reflective of those displayed by their nonlinear, GCN equivalents. It is typically observed that the attack efficacy is modestly reduced in the linear GNN setting, with further details deferred to Appendix D.2.

**Influence of Network Depth and Training Dynamics:** Table 1 indicates that the post-training performance of SERA is frequently less effective compared to the scenarios with randomly initialized weights. This observation may be attributed to the notion that supervised training tends to adversely affect the conditioning of weight matrices relative to their initialized state. Additionally, augmenting model depth does not correspond with enhanced attack efficacy, an outcome that is in alignment with our theoretical predictions.

## 7.2 Deficiency of SERA on dense graphs

**SBM experiments** In this section, we test SERA graph representations over SBM graphs with $K = 3, p = 0.3, q = 0.05$, with the rest of the experimental setups analogous to that in the Erdős–Rényi experiments. The evaluations are presented in figure 7. The results reveal the presence of a pronounced barrier that hinders the success of the attack across a wide range of configurations corresponding to different network depths and feature dimensions. Furthermore, we observe that the results tend to stabilize as the size of the graph increases. We provide a further study on the impact of SBM structure in appendix D.1.

## 7.3 Mitigation of SERA through NAG

In this section, we empirically study the defensive performance of noisy aggregation (8) against SERA We will use the Planetoid datasets [41] for evaluation. We consider a transductive node classification setting and use the standard train-test splits. The GNN models are trained using the training labels and evaluated on the test nodes. The performances of SERA are evaluated on the subgraphs induced by the test nodes. We report the configuration of GNN encoding, as well as the attacking pipeline and training hyperparameters in appendix D.3.1. Due to space limits, we report results on the Cora dataset under GCN and GAT in the main text and postpone the complete report in appendix D.3. We use the following two types of training configurations as proposed in section 6:

**Under the unconstrained scheme**, we use aggressive perturbation plans by applying noise with scale range $\sigma \in \{0, 1, 2, 4\}$, with $\sigma = 0$ indicating no protection, and $d \in \{2^i, 5 \leq i \leq 13\}$.

**Under the constrained scheme**, we adopt the spectral normalization technique [25] to control the spectral norm of each layer at approximately 1 (with relative error $< 10\%$). We use conservative perturbation plans by applying noise with scale range $\sigma \in \{0, 0.01, 0.05, 0.1, 0.5, 1\}$, and $d \in \{2^i, 5 \leq i \leq 13\}$. Note that with $\sigma = 1$, we obtain a non-vacuous lower bound according to (9). We present the evaluations in figure 2 and summarize our observations and findings as follows:

**SERA empirically elicits privacy-utility trade-off under the constrained scheme** When the noise level is moderate, i.e., $\sigma \in \{0.01, 0.05\}$. The result demonstrates that privacy and utility are, at least to some extent, at odds: Under lower noise level, SERA is able to achieve non-trivial success especially when $d$ is small. Furthermore, raising the feature dimension $d$ results in both a decrease

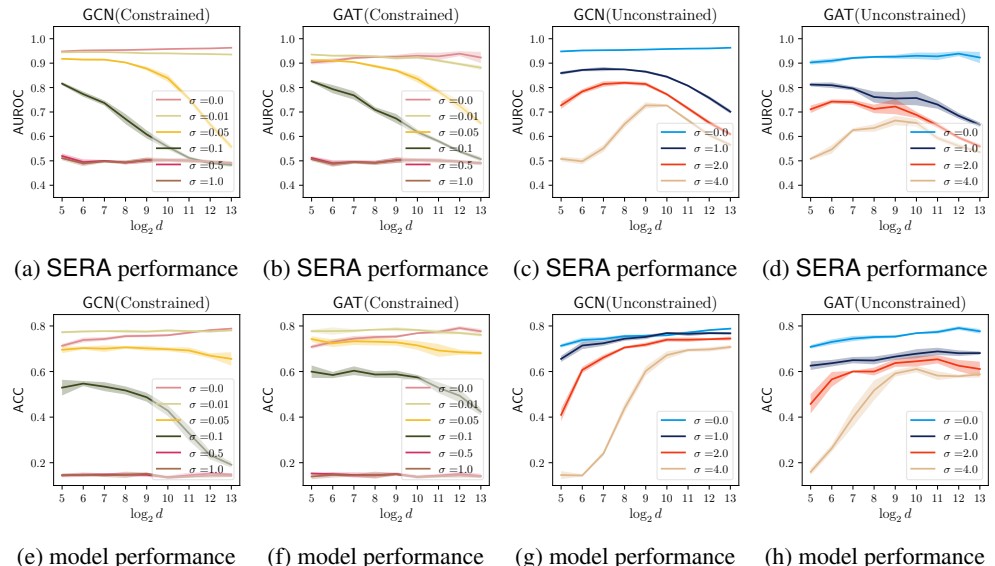

Figure 2: Privacy and utility assessments on the Cora dataset with underlying model of NAG being GCN and GAT. The first row contains attack performances of SERA measured using AUROC metric under both constrained and unconstrained training scheme. The second row presents corresponding model performances.

in utility as well as an increase in privacy. This is actually predictable: Since we explicitly control the operator norm to be around 1, a larger $d$ implies a smaller "signal-to-noise ratio" with the signal being (loosely) defined as the magnitude of the aggregated node representations.

**SERA losses power against NAG using larger $d$s in the unconstrained scheme** A surprising evidence according to figure 2 is that when the feature dimension $d$ is sufficiently raised, i.e., $d > 1024$, the attacking performances degrade. Consequently, we are able to achieve decent protection against SERA (AUROC $< 0.6$) while at the same time incurring slight degradation in model utility ($> 0.7$ Accuracy in Cora) Moreover, the phenomenon is more evident for higher noise levels. While the outcome seems favorable insofar as we have identified GNN solutions that manifest both high performance and a degree of privacy since the training procedure is unrelated to the attacking mechanism, these solutions may exhibit diminished robustness, as the corresponding Lipschitz constants are likely to be inadequately regulated [40]. Due to space limits, we postpone a more detailed discussion to appendix D.4.

## 8 Discussion and conclusion

In this paper, we have studied the behavior of the SERA adversary by characterizing its performance against different kinds of underlying graph structures as well as encoding mechanisms. Theoretically, we first identify sparse random graphs where SERA provably reconstructs the input graph, which ascertains the empirical findings of previous works. We then reveal limitations of SERA by showing its performance lower bounds when the input graph follows a dense SBM. Additionally, we discuss protection mechanisms to SERA by exploiting both theoretically and empirically the defensive capability of NAG. Empirical investigations corroborate with our theoretical findings, while suggesting intriguing phenomenons that questions the viability of SERA as a formal privacy auditing procedure for private graph representations. Notwithstanding, several research problems warrant further study, which we discuss in appendix E alongside with the limitations of this paper.

# 9 Acknowledgements

The authors from Ant Group are supported by the Leading Innovative and Entrepreneur Team Introduction Program of Hangzhou (Grant No.TD2022005). Guanhua Fang is partly supported by the National Natural Science Foundation of China (nos. 12301376).

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

# Appendix

## Table of Contents

## A   Broader impacts

The pervasive integration of graph representation learning (GRL) into various sectors, from social networks to bioinformatics, underscores the necessity of addressing the security and privacy risks inherent in these technologies. This paper contributes to the understanding of such risks by dissecting the structural vulnerabilities of graph representations under cosine-similarity-based edge reconstruction attacks (SERA). Our work has significant ethical implications and societal consequences, as we aim to balance the need for advanced data analytics with the imperative of safeguarding individual and community privacy.

Theoretically articulating the success and failure modes of SERA, our research offers a framework for evaluating GRL models against potential privacy breaches. The insights gained can guide the development of more secure algorithms that resist inadvertent information disclosure. By highlighting the efficacy of SERA in various settings, this paper also underscores the potential for such attacks to serve as auditing tools for privacy-preserving mechanisms, thereby fostering the creation of more trustworthy GRL systems.

As GRL technologies continue to evolve, our work calls attention to the importance of proactive privacy research in the field. It encourages the industry to adopt privacy-by-design principles and serves as a reminder to policymakers to consider the implications of GRL in legislation around data protection. Future societal consequences hinge on our ability to reconcile the benefits of GRL with the privacy rights of individuals, necessitating ongoing research, transparent practices, and informed governance to navigate this complex landscape.

# B  Additional backgrounds

## B.1  VFL and the SERA adversary

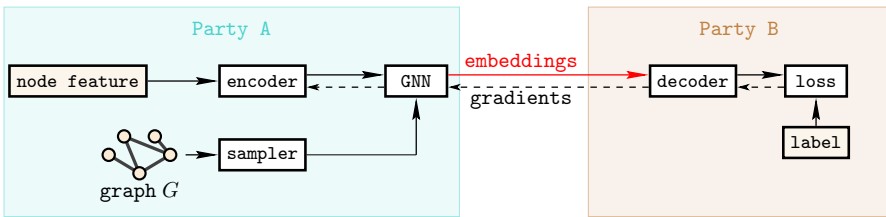

Figure 3: Illustration of a typical vertically federated graph representation learning scenario, the figure is adapted from [36].

We describe the scenario of vertically federated graph representation learning mentioned in section 1 in more detail, with the system architecture illustrated in figure 3.

**VFL setup** Under this scenario, we assume that party A (on the left side of figure 3 holds the graph as well as the node features, and party B (on the right side of figure 3 holds the node labels. Such kind of scenarios are encountered in applications like financial risk management [36]. Under VFL protocols, party A and party B iteratively exchange intermediate outputs to facilitate collaborative learning. The most representative method is split learning [36], where in each step, party A sends the node representations of a possibly sampled subgraph encoded using a graph neural network whose parameters are stored at party A. We call this operation the **forward transmission step** and highlighted in figure 3 as the red arrow.

**SERA adversary** We assume that party B is honest-but-curious in the sense that party B strictly follows the VFL protocol but tries to infer the graph structure belonged to party A, both during training and during inference, as the forward step is required for both stages. The attack requires nothing more than the VFL protocol: In each step, the two parties agree on a list of node indices that participates in this step (typically carried out using some cryptographically secure primitives), which constitutes the potential victim nodes $V_{\text{victim}}$. Upon receiving the node embeddings from party A, party B is then free to conduct SERA that targets the topological structure of $G_{\text{victim}}$. Furthermore, party B can even target a larger subgraph via storing multiple batches of embeddings and conduct attacks based on the unioned collections. Note that the adversary does not have access to GNN model parameters as they are kept locally at party A, which manifests the practicality of the proposed SERA adversary.

## B.2  Measures of homophily

The homophily metric is a way to describe the relationship between node properties and graph structure. Depending on the intrinsic nature of the property, we use two types of homophily measures: The label homophily (or edge homophily)

$$\mathcal{H}_{\text{label}}(G, Y) = \frac{1}{|E|} \sum_{(u,v) \in E} \mathbf{1}\left(Y_u = Y_v\right) \tag{10}$$

which measures the averaged agreement of adjacent nodes' labels and the feature homophily (or generalized homophily [18, 22])

$$\mathcal{H}_{\text{feature}}(G, X) = \frac{1}{|E|} \sum_{(u,v) \in E} \cos\left(X_u, X_v\right) \tag{11}$$

which replaces the agreement measure in the definition of label homophily with the cosine similarity between adjacent nodes' features. Another important metric we use in the experiments is the AUROC of guessing edge presence using feature similarities, i.e., $\widehat{A}_{uv}^{\text{FS}}(\tau) = \mathbf{1}\left(\cos(x_u, x_v) \geq \tau\right)$. Note that $\mathcal{H}_{\text{feature}}$ might be related to but not always correlate well with the AUROC of $\widehat{A}^{\text{FS}}$, as $\mathcal{H}_{\text{feature}}$ ignores the feature similarity of non-edges.

The feature homophily metric is sometimes related to, but not always correlated with the

### B.3  Representative GNNs and their corresponding aggregations rules

Here we briefly review GNN architectures that are involved in theorem 6.1 in the message-passing form as in (8):

**Mean pooling [14]**  This is the most standard form of message passing GNN. With the un-normalized and un-perturbed version analyzed in section 4 and 5:

$$H_v^{(l)} = \mathsf{ReLU}\left(\frac{1}{d_v+1}\sum_{u\in N(v)\cup\{v\}}\frac{W_l H_u^{(l-1)}}{\left\|H_u^{(l-1)}\right\|_2} + \epsilon\right) \tag{SAGE-meanpool}$$

**Summation pooling [38]**  This is a simplified version of the GIN model which is also analyzed in [36]:

$$H_v^{(l)} = \mathsf{ReLU}\left(\sum_{u\in N(v)\cup\{v\}}\frac{W_l H_u^{(l-1)}}{\left\|H_u^{(l-1)}\right\|_2} + \epsilon\right) \tag{GIN}$$

**Max pooling [14]**  In its un-normalized and un-perturbed version, this corresponds to the mostly used SAGE model:

$$H_v^{(l)} = \mathsf{ReLU}\left(\max_{u\in N(v)\cup\{v\}}\frac{W_l H_u^{(l-1)}}{\left\|H_u^{(l-1)}\right\|_2} + \epsilon\right) \tag{SAGE-maxpool}$$

**GCN pooling [20]**  The GCN pooling takes the form

$$H_v^{(l)} = \mathsf{ReLU}\left(\frac{1}{\sqrt{d_v+1}}\sum_{u\in N(v)\cup\{v\}}\frac{W_l H_u^{(l-1)}}{\sqrt{d_u+1}\left\|H_u^{(l-1)}\right\|_2} + \epsilon\right) \tag{GCN}$$

**Attentive pooling [32]**  This is also know as the GAT model. To simplify notations, let $\tilde{H}_v^{(l)} = H_v^{(l)}/\left\|H_v^{(l)}\right\|_2$, then the GAT model is recursively defined as

$$H_v^{(l)} = \mathsf{ReLU}\left(\sum_{u\in N(v)\cup\{v\}}\alpha_{uv}W_l\tilde{H}_u^{(l-1)} + \epsilon\right)$$

$$\alpha_{uv} = \frac{\exp\left(\mathsf{LeakyReLU}\left(\langle\beta_{\mathrm{src}},W_l\tilde{H}_u^{(l-1)}\rangle + \langle\beta_{\mathrm{dst}},W_l\tilde{H}_v^{(l-1)}\rangle\right)\right)}{\sum_{u\in N(v)\cup\{v\}}\exp\left(\mathsf{LeakyReLU}\left(\langle\beta_{\mathrm{src}},W_l\tilde{H}_v^{(l-1)}\rangle + \langle\beta_{\mathrm{dst}},W_l\tilde{H}_v^{(l-1)}\rangle\right)\right)} \tag{GAT}$$

where $\beta_{\mathrm{src}},\beta_{\mathrm{dst}}\in\mathbb{R}^d$ are learnable vector parameters.

## C  Proofs of Theorems

### C.1  Proof of theorem 4.1

In the proof, for notational simplicity, we abuse notation by treating $A = A + I$ and $D = D + I$ (i.e., self-edge is included in the edge graph). We then define $A^{(L)} := A \cdot \underbrace{...}_{L \text{ times}} \cdot A$ and $p_{ij}^{(L)} := ((D^{-1}A)^L)_{ij}$.

*Proof of the theorem.*  For any pair of two nodes $i$ and $j$, we next recall the formula of cosine similarity, $\cos\theta(H_i^{(L)}, H_j^{(L)})$,

$$\cos\theta(H_i^{(L)}, H_j^{(L)}) := \frac{\langle H_i^{(L)}, H_j^{(L)}\rangle}{\sqrt{\langle H_i^{(L)}, H_i^{(L)}\rangle \cdot \langle H_j^{(L)}, H_j^{(L)}\rangle}}, \tag{12}$$

which will be used recurrently in the following main proof.

According to the generation mechanism of node features (i.e., isotropic Gaussian assumption), we have that

$$|\frac{1}{d}\|X_j\|^2 - 1| \leq 3\sqrt{\frac{\log n}{d}} \text{ and } |\frac{1}{d}\langle X_j, X_{j'}\rangle| \leq 3\sqrt{\frac{\log n}{d}} \tag{13}$$

for all $j, j'$ with probability at least $1 - 1/n^2$.

*Case 1: without considering the learnable weight matrix $W$.* For the numerator in $\cos\theta(H_{i_1}^{(L)}, H_{i_2}^{(L)})$, when $i_1$ and $i_2$ are truly connected, we have

$$
\begin{aligned}
\langle H_{i_1}^{(L)}, H_{i_2}^{(L)}\rangle &= \sum_{j=1}^{n} p_{i_1 j}^{(L)} p_{i_2 j}^{(L)} \|X_j\|^2 + \sum_{j\neq j'} p_{i_1 j}^{(L)} p_{i_2 j'}^{(L)} \langle X_j, X_{j'}\rangle \\
&\geq p_{i_1 i_1}^{(L)} p_{i_2 i_1}^{(L)} \|X_{i_1}\|^2 + \sum_{j\neq j'} p_{i_1 j}^{(L)} p_{i_2 j}^{(L)} \langle X_j, X_{j'}\rangle \\
&\geq \frac{1}{|\mathcal{N}_{i_1}^{(L)}||\mathcal{N}_{i_2}^{(L)}|} \|X_{i_1}\|^2 + \sum_{j\neq j'} p_{i_1 j}^{(L)} p_{i_2 j}^{(L)} \langle X_j, X_{j'}\rangle \\
&\geq \frac{1}{(C_2 \log n)^{2L}} - 3\sqrt{\frac{\log n}{d}} \quad (\text{use the fact that } \sum_{j\neq j'} p_{i_1 j}^{(L)} p_{i_2 j'}^{(L)} \leq 1) \\
&\geq \frac{2}{3} \cdot \frac{1}{(C_2 \log n)^{2L}}
\end{aligned} \tag{14}
$$

when $d > 9(C_2 \log n)^{4L+2} \cdot \log n$. On the other hand, when $i_1$ and $i_2$ are not connected, by Lemma C.2, we know that, with high probability, there are at most $\frac{(C_2 \log n)^{2L}}{n} \cdot n(n-1)/2$ pairs of $i_1, i_2$ such that $\sum_j A_{i_1 j}^{(L)} A_{i_2 j}^{(L)} \geq 1$ which is equivalent to $\sum_j p_{i_1 j}^{(L)} p_{i_2 j}^{(L)} > 0$. For the rest of pairs, we have

$$
\begin{aligned}
\langle H_{i_1}^{(L)}, H_{i_2}^{(L)}\rangle &= \sum_{j\neq j'} p_{i_1 j}^{(L)} p_{i_2 j'}^{(L)} \langle X_j, X_{j'}\rangle \\
&\leq 3\sqrt{\frac{\log n}{d}} \\
&< \frac{1}{3} \cdot \frac{1}{(C_2 \log n)^{3L+1}},
\end{aligned} \tag{15}
$$

when $d > 9(C_2 \log n)^{6L+2} \cdot \log n$.

For the denominator $(\|H_{i_1}^{(L)}\| \cdot \|H_{i_2}^{(L)}\|)^{1/2}$, we give the upper and lower bounds of $\|H_i^{(L)}\|$. We can compute

$$
\begin{aligned}
\langle H_i^{(L)}, H_i^{(L)}\rangle &= \sum_{j=1}^{n} p_{ij}^{(L)} p_{ij}^{(L)} \|X_j\|^2 + \sum_{j\neq j'} p_{ij}^{(L)} p_{ij'}^{(L)} \langle X_j, X_{j'}\rangle \\
&\leq 1 + 3\sqrt{\frac{\log n}{d}} \\
&< 1 + \frac{1}{3} \cdot \frac{1}{(C_2 \log n)^{2L+1}},
\end{aligned} \tag{16}
$$

where we use the fact that $\sum_j p_{ij}^{(L)} p_{ij}^{(L)} \leq 1$. Conversely, we have

$$
\begin{aligned}
\langle H_i^{(L)}, H_i^{(L)}\rangle &= \sum_{j=1}^{n} p_{ij}^{(L)} p_{ij}^{(L)} \|X_j\|^2 + \sum_{j\neq j'} p_{ij}^{(L)} p_{ij'}^{(L)} \langle X_j, X_{j'}\rangle \\
&\geq \frac{1}{(C_2 \log n)^L} - 3\sqrt{\frac{\log n}{d}} \\
&\geq \frac{1}{(C_2 \log n)^L} - \frac{1}{3} \cdot \frac{1}{(C_2 \log n)^{2L+1}},
\end{aligned} \tag{17}
$$

where we use the fact that $\sum_j p_{ij}^{(L)} p_{ij}^{(L)} \geq 1/(C_2 \log n)^L$ when $|N_i^{(L)}| \leq (C_2 \log n)^L$.

To sum up, $\cos \theta(H_{i_1}^{(L)}, H_{i_2}^{(L)})$ is at least

$$\frac{2}{3} \cdot \frac{1}{(C_2 \log n)^{2L}} / (1 + \frac{1}{3(C_2 \log n)^{2L+1}}) \geq \frac{1}{2} \cdot \frac{1}{(C_2 \log n)^{2L}} \tag{18}$$

when node $i_1$ and $i_2$ are connected. On the other hand, $\cos \theta(H_{i_1}^{(L)}, H_{i_2}^{(L)})$ is at most

$$\frac{1}{3} \cdot \frac{1}{(C_2 \log n)^{3L+1}} / (\frac{1}{(C_2 \log n)^L} - \frac{1}{3} \cdot \frac{1}{(C_2 \log n)^{2L+1}}) < \frac{1}{2} \cdot \frac{1}{(C_2 \log n)^{2L}} \tag{19}$$

for all pairs (except at most $\frac{(C_2 \log n)^{2L}}{n} \cdot n(n-1)/2$ pairs) of disconnected nodes $i_1$ and $i_2$.

By choosing the cutoff $\tau = \frac{1}{2} \cdot \frac{1}{(C_2 \log n)^{2L}}$, with probability at least $1 - 2/n^2$, we have the false negative is zero and the false positive is $\frac{(C_2 \log n)^{2L}}{n}$.

*Case 2: with considering the learnable weight matrix $W$.* Additionally, if the learnable weight $W$ is taken into account, we can derive the following results. We define $\kappa_1$ and $\kappa_2$ to be the largest and smallest positive constants such that

$$\kappa_1 \langle X, X' \rangle \leq \langle WX, WX' \rangle \leq \kappa_2 \langle X, X' \rangle$$

holds. It is easy to see that $\kappa_2/\kappa_1 = (\kappa(W))^2$. Then the parallel version of (14) becomes

$$\langle H_{i_1}^{(L)}, H_{i_2}^{(L)} \rangle \geq \kappa_1 \frac{2}{3} \frac{1}{(C_2 \log n)^{2L}}. \tag{20}$$

The parallel version of (15) becomes

$$\langle H_{i_1}^{(L)}, H_{i_2}^{(L)} \rangle \leq 3\kappa_2 \sqrt{\frac{\log n}{d}}. \tag{21}$$

The parallel version of (16) becomes

$$\langle H_i^{(L)}, H_i^{(L)} \rangle \leq \kappa_2 (1 + 3\sqrt{\frac{\log n}{d}}). \tag{22}$$

The parallel version of (17) becomes

$$\langle H_{i_1}^{(L)}, H_{i_2}^{(L)} \rangle \geq \kappa_1 (\frac{1}{(C_2 \log n)^L} - 3\sqrt{\frac{\log n}{d}}). \tag{23}$$

Combining above results, we have that

$$\cos \theta(H_{i_1}^{(L)}, H_{i_2}^{(L)}) \geq \frac{\kappa_1 \frac{2}{3} \frac{1}{(C_2 \log n)^{2L}}}{\kappa_2 (1 + 3\sqrt{\frac{\log n}{d}})}$$

$$\geq \frac{\kappa_1}{2\kappa_2} \frac{1}{(C_2 \log n)^{2L}} =: \mathrm{cut}_1(L) \tag{24}$$

when $i_1, i_2$ are connected and $d \gg \log^2 n$ and

$$\cos \theta(H_{i_1}^{(L)}, H_{i_2}^{(L)}) \leq \frac{3\kappa_2 \sqrt{\frac{\log n}{d}}}{\kappa_1 (\frac{1}{(C_2 \log n)^L} - 3\sqrt{\frac{\log n}{d}})}$$

$$\leq \frac{4\kappa_2}{\kappa_1} \frac{\sqrt{\frac{\log n}{d}}}{\frac{1}{(C_2 \log n)^L}} =: \mathrm{cut}_2(L) \tag{25}$$

when $i_1, i_2$ are not connected and $d \gg (C_2 \log n)^{6L+2} \log n$.

Therefore as long as $d \gg (C_2 \log n)^{6L+2} \log n$ and

$$(\frac{\kappa_1}{\kappa_2})^2 \geq 8(C_2 \log n)^{3L} \cdot \sqrt{\frac{\log n}{d}} \tag{26}$$

holds, we can choose any cutoff $\tau$ between $\mathrm{cut}_1(L)$ and $\mathrm{cut}_2(L)$ so that false negative rate is zero and false positive rate is no larger than $(C_2 \log n)^{2L}/n$. This completes the proof regarding FPR and FNR. For the implications in AUROC, the result follows immediately by noting the discoveries are montone in $\tau$.

**Supporting Lemma of Theorem 4.1**    The following Lemmas are used for controlling the number of pairs of nodes $(u, v)$'s which satisfy $\sum_j A_{uj}^{(L)} A_{vj}^{(L)} \geq 1$.

**Lemma C.1.** *Let $B(n, p)$ denote the binomial distribution with probability $p$ and size $n$.*

1. *Suppose $X$ dominates $B(n, p)$. For any $a > 0$, we have*

$$\mathbb{P}(X < np - a) \leq \exp\{-a^2/2np\}. \tag{27}$$

2. *Suppose $X$ is dominated by $B(n, p)$. For any $a > 0$, we have*

$$\mathbb{P}(X > np + a) \leq \min\{\exp\{-a^2/2np + a^3/(np)^3\}, \exp\{-\frac{a^2}{2np + 2a/3}\}\}. \tag{28}$$

Proof of Lemma C.1 is standard and we omit it here. The consequences of this Lemma is that $\frac{C_1}{2} \log n \leq \sum_j A_{ij} \leq \frac{3C_1}{2} \log n$ with high probability at least $1 - 1/n^2$ for all node $i \in [n]$.

**Lemma C.2.** *Given a graph with edge probability $p$ ($p \leq C_1 \frac{\log n}{n}$), then*

$$\mathbb{P}(\sum_j A_{i_1 j}^{(L)} A_{i_2 j}^{(L)} \geq 1) \leq \frac{(C_2 \log n)^{2L}}{n - 1}, \tag{29}$$

*where $i_1, i_2$ are two nodes uniformly randomly sampled from the graph.*

*Proof of Lemma C.2.*  By recalling the definition of $A_{ij}^{(L)}$ that $A_{ij}^{(L)}$ equals one only when node $i$ and node $j$ can be connected within a path of length $L$. Therefore, with probability at least $1 - 1/n^2$, it holds $|\mathcal{N}_j^{(L)}| \leq (\frac{3C_1 \log n}{2})^L$, where $\mathcal{N}_j^{(L)} := \{i : A_{ij}^{(L)} = 1\}$

Note that, given fixed $j$, $A_{i_1 j} A_{i_2 j}$ is greater than 0 only if $i_1, i_2 \in \mathcal{N}_j^{(L)}$. By the symmetry, we know that this happens with probability at most $\frac{|\mathcal{N}_j^{(L)}|(|\mathcal{N}_j^{(L)}| - 1)}{n(n-1)}$ when $i_1, i_2$ are uniformly randomly sampled. Therefore, by union bound, we have

$$
\begin{aligned}
\mathbb{P}(\sum_j A_{i_1 j}^{(L)} A_{i_2 j}^{(L)} \geq 1) &\leq \sum_j \frac{|\mathcal{N}_j^{(L)}|(|\mathcal{N}_j^{(L)}| - 1)}{n(n-1)} \\
&\leq \frac{(1.5 C_1 \log n)^{2L}}{n - 1},
\end{aligned} \tag{30}
$$

which concludes the proof.    □

The implication of this lemma is that there are at most $n \cdot (C_2 \log n)^{2L}$ pairs of $(u, v)$ such that $\sum_j A_{uj}^{(L)} A_{vj}^{(L)} \geq 1$.

**Extension to different similarities**    The cosine similarity can be replaced by other similarity metrics. For example, we can use correlation similarity, i.e.,

$$\text{corr}(H_i^{(L)}, H_j^{(L)}) :=:= \frac{\langle H_i^{(L)} - \bar{H}_i^{(L)}, H_j^{(L)} - \bar{H}_j^{(L)} \rangle}{\sqrt{\langle H_i^{(L)} - \bar{H}_i^{(L)}, H_i^{(L)} - \bar{H}_i^{(L)} \rangle \cdot \langle H_j^{(L)} - \bar{H}_j^{(L)}, H_j^{(L)} - \bar{H}_j^{(L)} \rangle}},$$

where $\bar{H}_i^{(L)}$ is a $d$-dimensional vector whose elements are equal to the mean of $H_i^{(L)}$. By treating $X_j - \bar{X}_j$ as $X_j$ ($j \in [n]$), where $\bar{X}_j$ is a $d$-dimensional vector whose elements are equal to the mean of $X_j$. (13) changes to

$$|\frac{1}{d} \|X_j\|^2 - 1| \leq 4\sqrt{\frac{\log n}{d}} \quad \text{and} \quad |\frac{1}{d} \langle X_j, X_{j'} \rangle| \leq 4\sqrt{\frac{\log n}{d}}.$$

Therefore, the above proof still holds by adjusting the constant accordingly.

## C.2 Proof of theorem 5.1

To prove the desired result, we first need the following lemmas. In the rest of proof, we abuse the notation by treating $p$ as $p_0$ and $q$ as $q_0$.

By applying the Hoeffding's inequality, we can obtain the following two lemmas.

**Lemma C.3.** *It holds* $|\sum_{j:i,j \text{ in the same group}} A_{ij} - \frac{n}{K} \cdot p_0| \le 3 \log n =: \epsilon_1$ *for all $i$ with probability at least $1 - 1/n^2$.*

**Lemma C.4.** *Suppose $i$ is in group $k$, it holds* $|\sum_{j:i,j \text{ in the group } k'(\ne k)} A_{ij} - \frac{n}{K} \cdot q_0| \le \min\{\frac{1}{2}\frac{n}{K} \cdot q_0, 3 \log n\} =: \epsilon_2$ *for all $i$ with probability at least $1 - 1/n^2$.*

Combining Lemma C.3 and Lemma C.4, we have the following lemma.

**Lemma C.5.** *It holds* $|\sum_j A_{ij} - (\frac{n}{K} \cdot p_0 + (n - \frac{n}{K}) \cdot q_0)| \le \epsilon_1 + (K-1)\epsilon_2$ *with probability at least $1 - 2/n^2$.*

In summary, with high probability confidence, Lemma C.5 gives the characterization of degree (i.e. number of neighbours) of every node $i$.

We then make a step forward and characterize the normalized degree $p_{ij}^{(L)}$ for $L \ge 2$ in the following lemmas.

**Lemma C.6.** *With probability at least $1 - 1/n^2$, it holds that* $|A_{ij}^{(2)} - (\frac{n}{K}p_0^2 + (n - n/K)q_0 p_0)| \le 6 \log n + \frac{1}{2}\frac{n}{K} \cdot q_0$ *for $i, j$ from the same group and* $|A_{ij}^{(2)} - (\frac{n}{K}p_0 q_0 + (n - n/K)q_0^2)| \le \min\{\frac{2}{3}(\frac{n}{K}p_0 q_0 + (n - n/K)q_0^2), 3(K-1)\log n\}$ *for $i, j$ from different groups.*

**Lemma C.7.** *For $L \ge 2$, suppose there exist constants $a_1^{(L)}$ and $a_2^{(L)}$ such that $|A_{ij}^{(L)} - a_1^{(L)}| \le \epsilon_1^{(L)}$ when $i, j$ are in the same group and $|A_{ij}^{(L)} - a_2^{(L)}| \le \epsilon_2^{(L)}$ when $i, j$ are not in the same group. It holds that*

$$|A_{ij}^{(L+1)} - a_1^{(L+1)}| \le \epsilon_1^{(L)} \quad i, j \text{ in the same group}$$
$$|A_{ij}^{(L+1)} - a_2^{(L+1)}| \le \epsilon_2^{(L)} \quad i, j \text{ not in the same group}, \tag{31}$$

*with*

$$a_1^{(L+1)} := (a_1^{(L)} \frac{n}{K} p_0 + a_2^{(L)}(n - n/K)q_0),$$
$$a_2^{(L+1)} := a_1^{(L)} \frac{n}{K} q_0 + a_2^{(L)} \frac{n}{K} p_0 + a_2^{(L)}(n - 2n/K)q_0$$
$$\epsilon_1^{(L+1)} := \epsilon_1^{(L)} \frac{n}{K} p_0 + \epsilon_1 a_1^{(L)} + \epsilon_1 \epsilon_1^{(L)} + \epsilon_2^{(L)}(n - n/K)q_0 + (K-1)\epsilon_2 a_2^{(L)} + (K-1)\epsilon_2 \epsilon_2^{(L)},$$
$$\epsilon_2^{(L+1)} := \epsilon_1^{(L)} \frac{n}{K} q_0 + \epsilon_2 a_1^{(L)} + \epsilon_2 \epsilon_1^{(L)} + \epsilon_2^{(L)} \frac{n}{K} p_0 + \epsilon_1 a_2^{(L)} + \epsilon_1 \epsilon_2^{(L)} + \epsilon_2^{(L)}(n - 2n/K)q_0$$
$$+ (K-2)\epsilon_2 a_2^{(L)} + (K-2)\epsilon_2 \epsilon_2^{(L)}.$$

Proof of Lemma C.6 is a special case of that of Lemma C.7. In the following, we prove Lemma C.7.

*Proof of Lemma C.7.* By the definition, we know $A_{ij}^{(L+1)} = \sum_{j'} A_{ij'}^{(L)} A_{j'j}$.

When $i, j$ are from the same class (w.l.o.g, we denote it as class 1), then it holds

$$|A_{ij}^{(L+1)} - (a_1^{(L)} \frac{n}{K} p_0 + a_2^{(L)}(n - n/K)q_0)|$$
$$= |\sum_{j'} A_{ij'}^{(L)} A_{j'j} - (a_1^{(L)} \frac{n}{K} p_0 + a_2^{(L)}(n - n/K)q_0)|$$
$$\le |\sum_{j' \text{ in class 1}} A_{ij'}^{(L)} A_{j'j} - a_1^{(L)} \frac{n}{K} p_0| + |\sum_{j' \text{ not in class 1}} A_{ij'}^{(L)} A_{j'j} - a_2^{(L)}(n - n/K)q_0|$$
$$= \epsilon_1^{(L)} \frac{n}{K} p_0 + \epsilon_1 a_1^{(L)} + \epsilon_1 \epsilon_1^{(L)} + \epsilon_2^{(L)}(n - n/K)q_0 + (K-1)\epsilon_2 a_2^{(L)} + (K-1)\epsilon_2 \epsilon_2^{(L)} \tag{32}$$

Therefore, we can let $a_1^{(L+1)} := (a_1^{(L)} \frac{n}{k} p_0 + a_2^{(L)}(n - n/K)q_0)$ and $\epsilon_1^{(L+1)} := \epsilon_1^{(L)} \frac{n}{K} p_0 + \epsilon_1 a_1^{(L)} + \epsilon_1 \epsilon_1^{(L)} + \epsilon_2^{(L)}(n - n/K)q_0 + (K-1)\epsilon_2 a_2^{(L)} + (K-1)\epsilon_2 \epsilon_2^{(L)}$.

When $i, j$ are not from the same class (w.l.o.g. we assume $i$ is from class 1 and $j$ is from class 2), then it holds

$$
\begin{aligned}
&|A_{ij}^{(L+1)} - (a_1^{(L)} \frac{n}{K} q_0 + a_2^{(L)} \frac{n}{K} p_0 + a_2^{(L)}(n - 2n/K)q_0)| \\
=\quad &|\sum_{j'} A_{ij'}^{(L)} A_{j'j} - (a_1^{(L)} \frac{n}{K} q_0 + a_2^{(L)} \frac{n}{K} p_0 + a_2^{(L)}(n - 2n/K)q_0)| \\
\leq\quad &|\sum_{j' \text{ in class } 1} A_{ij'}^{(L)} A_{j'j} - a_1^{(L)} \frac{n}{k} q_0| + |\sum_{j' \text{ in class } 2} A_{ij'}^{(L)} A_{j'j} - a_2^{(L)} \frac{n}{K} p_0| \\
&+ |\sum_{j' \text{ not in class } 1 \& 2} A_{ij'}^{(L)} A_{j'j} - a_2^{(L)}(n - 2n/K)q_0| \\
\leq\quad &\epsilon_1^{(L)} \frac{n}{K} q_0 + \epsilon_2 a_1^{(L)} + \epsilon_2 \epsilon_1^{(L)} + \epsilon_2^{(L)} \frac{n}{K} p_0 + \epsilon_1 a_2^{(L)} + \epsilon_1 \epsilon_2^{(L)} \\
&+ \epsilon_2^{(L)}(n - 2n/K)q_0 + (K-2)\epsilon_2 a_2^{(L)} + (K-2)\epsilon_2 \epsilon_2^{(L)}. \quad (33)
\end{aligned}
$$

Therefore, we can let $a_2^{(L+1)} := a_1^{(L)} \frac{n}{K} q_0 + a_2^{(L)} \frac{n}{K} p_0 + a_2^{(L)}(n - 2n/K)q_0$ and $\epsilon_2^{(L+1)} := \epsilon_1^{(L)} \frac{n}{K} q_0 + \epsilon_2 a_1^{(L)} + \epsilon_2 \epsilon_1^{(L)} + \epsilon_2^{(L)} \frac{n}{K} p_0 + \epsilon_1 a_2^{(L)} + \epsilon_1 \epsilon_2^{(L)} + \epsilon_2^{(L)}(n - 2n/K)q_0 + (K-2)\epsilon_2 a_2^{(L)} + (K-2)\epsilon_2 \epsilon_2^{(L)}$. $\quad\square$

By above induction, it can be seen that, for any fixed $L$, $\epsilon_1^{(L)}/a_1^{(L)} = O_p(\frac{\log n}{n})$, $\epsilon_2^{(L)}/a_1^{(L)} = O_p(\frac{\log n}{n})$. It also holds $a_2^{(L)}/a_1^{(L)} = O_p(\frac{\log n}{n})$ when true edge probability satisfies $q_0 = O_p(\frac{\log n}{n})$, and $\epsilon_2^{(L)}/a_2^{(L)} = O_p(\frac{\log n}{n})$ when $q_0 \gg \frac{\log n}{n}$.

Recall the definition that $p_{ij}^{(L)} = ((D^{-1}A)^L)_{ij}$, therefore $p_{ij}^{(L)} \propto A_{ij}^{(L)}$ for any fixed $i$. In other words, for fixed $L \geq 2$, we have

$$
p_{ij}^{(L)} := \bar{p}_{ij}^{(L)} + O_p(\frac{k \log n}{n^2}) = \underbrace{\frac{a_1^{(L)}}{\frac{n}{K} \cdot a_1^{(L)} + (n - \frac{n}{K}) \cdot a_2^{(L)}}}_{=:p_1^{(L)}} + O_p(\frac{k \log n}{n^2}), \quad i, j \text{ in the same group,}
$$

$$
p_{ij}^{(L)} = \bar{p}_{ij}^{(L)} + O_p(\frac{k \log n}{n^2}) = \underbrace{\frac{a_2^{(L)}}{\frac{n}{K} \cdot a_1^{(L)} + (n - \frac{n}{K}) \cdot a_2^{(L)}}}_{=:p_2^{(L)}} + O_p(\frac{k \log n}{n^2}), \quad i, j \text{ not in the same group.}
$$

$$(34)$$

Here, on a very high level, we can treat $\bar{p}_{ij}^{(L)}$ as the population version of $((D^{-1}A)^L)_{ij}$. When $i, j$ in the same group, then $\bar{p}_{ij}^{(L)} \equiv p_1^{(L)}$. Otherwise, $\bar{p}_{ij}^{(L)} \equiv p_2^{(L)}$. With above preparations, we are ready to prove the theorem as follows.

*Proof of the theorem.* We need to consider the case $L \geq 2$ and $L = 1$ separately.

*Case 1: $L \geq 2$.* We define $\bar{X}_k := \sum_{i \in \text{ group k}} X_i$ and $r^{(L)} := a_2^{(L)}/a_1^{(L)}$. For the numerator in $\cos\theta(H_{i_1}^{(L)}, H_{i_2}^{(L)})$, when $i_1$ and $i_2$ are in the same group (w.l.o.g, suppose it is group 1), we have

$$
\begin{aligned}
\langle H_{i_1}^{(L)}, H_{i_2}^{(L)} \rangle &= \sum_{j=1}^{n} p_{i_1 j}^{(L)} p_{i_2 j}^{(L)} \|X_j\|^2 + \sum_{j \neq j'} p_{i_1 j}^{(L)} p_{i_2 j'}^{(L)} \langle X_j, X_{j'} \rangle \\
&= \sum_{j=1}^{n} \bar{p}_{i_1 j}^{(L)} \bar{p}_{i_2 j}^{(L)} \|X_j\|^2 + \sum_{j \neq j'} \bar{p}_{i_1 j}^{(L)} \bar{p}_{i_2 j'}^{(L)} \langle X_j, X_{j'} \rangle + \text{error} \qquad (35) \\
&= p_1^{(L)} p_1^{(L)} \langle \bar{X}_1, \bar{X}_1 \rangle + 2 \sum_{k \neq 1} p_1^{(L)} p_2^{(L)} \langle \bar{X}_1, \bar{X}_k \rangle + \sum_{k \neq 1} p_2^{(L)} p_2^{(L)} \langle \bar{X}_k, \bar{X}_k \rangle \\
&\quad + \sum_{k \neq k' \neq 1} p_2^{(L)} p_2^{(L)} \langle \bar{X}_k, \bar{X}_{k'} \rangle + \text{error} \\
&= p_1^{(L)} p_1^{(L)} \frac{n}{K} + (K-1) p_2^{(L)} p_2^{(L)} \frac{n}{K} + O_p\left((K-1) p_1^{(L)} p_2^{(L)} \frac{n}{K} \frac{1}{\sqrt{d}}\right) \qquad (36) \\
&\quad + (K-1)(K-2) p_2^{(L)} p_2^{(L)} \frac{n}{K} \frac{1}{\sqrt{d}}\Big) + \text{error},
\end{aligned}
$$

where (36) uses the property of node feature generation mechanism that $\langle \bar{X}_k, \bar{X}_k \rangle = \frac{n}{K}(1 + \sqrt{1/d})$ for any $k$ and $\langle \bar{X}_k, \bar{X}_{k'} \rangle = O_p(\frac{n}{K\sqrt{d}})$ for $k \neq k'$. Here the error term in (36) is error $:= \sum_{j=1}^{n} (p_{i_1 j}^{(L)} p_{i_2 j}^{(L)} - \bar{p}_{i_1 j}^{(L)} \bar{p}_{i_2 j}^{(L)}) \|X_j\|^2 + \sum_{j \neq j'} (p_{i_1 j}^{(L)} p_{i_2 j'}^{(L)} - \bar{p}_{i_1 j}^{(L)} \bar{p}_{i_2 j'}^{(L)}) \langle X_j, X_{j'} \rangle$, which can be controlled as follows.

$$
\begin{aligned}
|\text{error}| &= \Big| \sum_{j=1}^{n} (p_{i_1 j}^{(L)} p_{i_2 j}^{(L)} - \bar{p}_{i_1 j}^{(L)} \bar{p}_{i_2 j}^{(L)}) \|X_j\|^2 + \sum_{j \neq j'} (p_{i_1 j}^{(L)} p_{i_2 j'}^{(L)} - \bar{p}_{i_1 j}^{(L)} \bar{p}_{i_2 j'}^{(L)}) \langle X_j, X_{j'} \rangle \Big| \\
&\leq \Big| \sum_{j=1}^{n} (p_{i_1 j}^{(L)} p_{i_2 j}^{(L)} - \bar{p}_{i_1 j}^{(L)} \bar{p}_{i_2 j}^{(L)}) \|X_j\|^2 \Big| + \Big| \sum_{j \neq j'} (p_{i_1 j}^{(L)} p_{i_2 j'}^{(L)} - \bar{p}_{i_1 j}^{(L)} \bar{p}_{i_2 j'}^{(L)}) \langle X_j, X_{j'} \rangle \Big| \\
&\leq C\Big(\frac{k \log n}{n^2} + \sum_j \frac{k \log n}{n^2} \sqrt{\frac{\log n}{d}}\Big) \\
&= O_p\Big(\frac{k \log n}{n^2} + \frac{k \log n}{n} \sqrt{\frac{\log n}{d}}\Big), \qquad (37)
\end{aligned}
$$

where (37) utilizes the fact that $\sum_j \bar{p}_{ij} \equiv 1$ for any $i$ and (34) by adjusting the constant.

When $i_1, i_2$ are not in the same group (w.l.o.g, suppose $i_1$ in group 1 and $i_2$ in group 2), we have

$$
\begin{aligned}
\langle H_{i_1}^{(L)}, H_{i_2}^{(L)} \rangle &= \sum_{j=1}^{n} p_{i_1 j}^{(L)} p_{i_2 j}^{(L)} \|X_j\|^2 + \sum_{j \neq j'} p_{i_1 j}^{(L)} p_{i_2 j'}^{(L)} \langle X_j, X_{j'} \rangle \\
&= \sum_{j=1}^{n} \bar{p}_{i_1 j}^{(L)} \bar{p}_{i_2 j}^{(L)} \|X_j\|^2 + \sum_{j \neq j'} \bar{p}_{i_1 j}^{(L)} \bar{p}_{i_2 j'}^{(L)} \langle X_j, X_{j'} \rangle + \text{error} \\
&= p_1^{(L)} p_2^{(L)} (\langle \bar{X}_1, \bar{X}_1 \rangle + \langle \bar{X}_2, \bar{X}_2 \rangle) + (p_1^{(L)} p_1^{(L)} + p_2^{(L)} p_2^{(L)}) \langle \bar{X}_1, \bar{X}_2 \rangle \\
&\quad + (p_1^{(L)} p_2^{(L)} + p_2^{(L)} p_2^{(L)}) \sum_{k \neq 1,2} \langle \bar{X}_1 + \bar{X}_2, \bar{X}_k \rangle + \sum_{k \neq 1,2} p_2^{(L)} p_2^{(L)} \langle \bar{X}_k, \bar{X}_k \rangle \\
&\quad + \sum_{k \neq k' \neq 1,2} p_2^{(L)} p_2^{(L)} \langle \bar{X}_k, \bar{X}_{k'} \rangle + \text{error} \\
&= 2 p_1^{(L)} p_2^{(L)} \frac{n}{K} + (K-2) p_2^{(L)} p_2^{(L)} \frac{n}{K} \\
&\quad + O_p\Big((p_1^{(L)} p_1^{(L)} + K p_1^{(L)} p_2^{(L)} + K^2 p_2^{(L)} p_2^{(L)}) \frac{n}{K} \sqrt{\frac{1}{d}}\Big) + \text{error}. \qquad (38)
\end{aligned}
$$

To sum up, if $i_1, i_2$ are in the same group, $\cos\theta(H_{i_1}^{(L)}, H_{i_2}^{(L)})$ satisfies

$$
\begin{aligned}
&\cos\theta(H_{i_1}^{(L)}, H_{i_2}^{(L)}) \\
&= \frac{\langle H_{i_1}^{(L)}, H_{i_2}^{(L)} \rangle}{\sqrt{\langle H_{i_1}^{(L)}, H_{i_1}^{(L)} \rangle \cdot \langle H_{i_2}^{(L)}, H_{i_2}^{(L)} \rangle}} \\
&= \frac{p_1^{(L)} p_1^{(L)} \frac{n}{k} + (K-1) p_2^{(L)} p_2^{(L)} \frac{n}{K}}{p_1^{(L)} p_1^{(L)} \frac{n}{K} + (K-1) p_2^{(L)} p_2^{(L)} \frac{n}{K} + C\left((K p_1^{(L)} p_2^{(L)} + K^2 p_2^{(L)2}) \frac{n}{K\sqrt{d}} + O_p(\frac{K\log n}{n}(\frac{1}{n} + \sqrt{\frac{\log n}{d}}))\right)} \\
&= \underbrace{1}_{\mathrm{cut}_1(L)} - o_p(1)
\end{aligned}
\tag{39}
$$

as long as $d \gg K^2 \log^3 n / b^2$.

If $i_1, i_2$ are not in the same group, $\cos\theta(H_{i_1}^{(L)}, H_{i_2}^{(L)})$ satisfies

$$
\begin{aligned}
&\cos\theta(H_{i_1}^{(L)}, H_{i_2}^{(L)}) \\
&= \frac{\langle H_{i_1}^{(L)}, H_{i_2}^{(L)} \rangle}{\sqrt{\langle H_{i_1}^{(L)}, H_{i_1}^{(L)} \rangle \cdot \langle H_{i_2}^{(L)}, H_{i_2}^{(L)} \rangle}} \\
&= \frac{2 p_1^{(L)} p_2^{(L)} \frac{n}{K} + (K-2) p_2^{(L)} p_2^{(L)} \frac{n}{K} + C\left((K p_1^{(L)} p_2^{(L)} + K^2 p_2^{(L)2}) \frac{n}{K\sqrt{d}} + O_p(\frac{K\log n}{n}(\frac{1}{n} + \sqrt{\frac{\log n}{d}}))\right)}{p_1^{(L)} p_1^{(L)} \frac{n}{K} + (K-1) p_2^{(L)} p_2^{(L)} \frac{n}{K}} \\
&= \underbrace{\frac{2 r^{(L)} + (K-2) r^{(L)2}}{1 + (K-1) r^{(L)2}}}_{\mathrm{cut}_2(L)} + o_p(1).
\end{aligned}
\tag{40}
$$

Remark. As $L \to \infty$, $r^{(L)}$ will converge to 1. Therefore, $\mathrm{cut}_2(L)$ will eventually equal $1 \equiv \mathrm{cut}_1(L)$.

*Case 2: $L = 1$.* For notational convenience, we define $\tilde{X}_{k,1}^{(i)} := \sum_{i\in \text{ group } k} b_{i,1}^{(i)} X_i$ where $b_{i,1}^{(i)}$'s are i.i.d. Bernoulli random variables with success probability $p_0$ and $\tilde{X}_{k,2}^{(i)} := \sum_{i\in \text{ group } k} b_{i,2}^{(i)} X_i$ where $b_{i,2}^{(i)}$'s are i.i.d. Bernoulli random variables with success probability $q_0$.

Then it is straightforward to calculate that, if $i_1, i_2$ are in the same group 1, it holds

$$
\begin{aligned}
\langle H_{i_1}^{(1)}, H_{i_2}^{(1)} \rangle \quad =_d \quad & \frac{1}{D_{i_1} D_{i_2}} \Big( \langle \tilde{X}_{1,1}^{(i_1)}, \tilde{X}_{1,1}^{(i_2)} \rangle + \sum_{k\neq 1} \langle \tilde{X}_{1,1}^{(i_1)}, \tilde{X}_{k,2}^{(i_2)} \rangle + \sum_{k\neq 1} \langle \tilde{X}_{k,2}^{(i_1)}, \tilde{X}_{1,1}^{(i_2)} \rangle \\
& + \sum_{k\neq 1} \langle \tilde{X}_{k,2}^{(i_1)}, \tilde{X}_{k,2}^{(i_2)} \rangle + \sum_{k,k'\neq 1} \langle \tilde{X}_{k,2}^{(i_1)}, \tilde{X}_{k',2}^{(i_2)} \rangle \Big) \\
= \quad & \frac{1}{D_{i_1} D_{i_2}} \Big( \frac{n}{k} p_0^2 + (K-1) \frac{n}{K} q_0^2 + O_p\big(p_0 \frac{n\sqrt{\log n}}{K\sqrt{d}} + \sqrt{p_0 q_0} \frac{n\sqrt{\log n}}{\sqrt{d}} \\
& + K q_0 \frac{n\sqrt{\log n}}{\sqrt{d}} \big) \Big).
\end{aligned}
\tag{41}
$$

Similarly, if $i_1, i_2$ are not in the same group (w.l.o.g, they are in group 1 and 2 respectively), it holds

$$
\begin{aligned}
\langle H_{i_1}^{(1)}, H_{i_2}^{(1)} \rangle \ =_d \ & \frac{1}{D_{i_1} D_{i_2}} \big( \langle \tilde{X}_{1,1}^{(i_1)}, \tilde{X}_{1,2}^{(i_2)} \rangle + \langle \tilde{X}_{2,2}^{(i_1)}, \tilde{X}_{1,1}^{(i_2)} \rangle + \langle \tilde{X}_{1,1}^{(i_1)}, \tilde{X}_{2,1}^{(i_2)} \rangle + \langle \tilde{X}_{2,2}^{(i_1)}, \tilde{X}_{1,2}^{(i_2)} \rangle \\
& + \sum_{k \neq 1,2} \langle \tilde{X}_{1,1}^{(i_1)} + \tilde{X}_{2,2}^{(i_1)}, \tilde{X}_{k,2}^{(i_2)} \rangle + \sum_{k \neq 1,2} \langle \tilde{X}_{k,2}^{(i_1)}, \tilde{X}_{1,2}^{(i_2)} + \tilde{X}_{2,1}^{(i_2)} \rangle \\
& + \sum_{k \neq 1,2} \langle \tilde{X}_{k,2}^{(i_1)}, \tilde{X}_{k,2}^{(i_2)} \rangle + \sum_{k,k' \neq 1,2} \langle \tilde{X}_{k,2}^{(i_1)}, \tilde{X}_{k',2}^{(i_2)} \rangle \big) \\
= \ & \frac{1}{D_{i_1} D_{i_2}} \big( 2 \frac{n}{k} p_0 q_0 + (K-2) \frac{n}{K} q_0^2 + O_p (p_0 \frac{n\sqrt{\log n}}{K\sqrt{d}} + \sqrt{p_0 q_0} \frac{n\sqrt{\log n}}{\sqrt{d}} \\
& + K q_0 \frac{n\sqrt{\log n}}{\sqrt{d}})\big).
\end{aligned}
\tag{42}
$$

Moreover, if $i_1 = i_2$, it holds

$$
\begin{aligned}
\langle H_{i_1}^{(1)}, H_{i_1}^{(1)} \rangle \ =_d \ & \frac{1}{D_{i_1} D_{i_1}} \big( \langle \tilde{X}_{1,1}^{(i_1)}, \tilde{X}_{1,1}^{(i_1)} \rangle + 2 \sum_{k \neq 1} \langle \tilde{X}_{1,1}^{(i_1)}, \tilde{X}_{k,2}^{(i_1)} \rangle \big) \\
& + \sum_{k \neq 1} \langle \tilde{X}_{k,2}^{(i_1)}, \tilde{X}_{k,2}^{(i_1)} \rangle + \sum_{k,k' \neq 1} \langle \tilde{X}_{k,2}^{(i_1)}, \tilde{X}_{k',2}^{(i_1)} \rangle \big) \\
= \ & \frac{1}{D_{i_1} D_{i_2}} \big( \frac{n}{k} p_0 + (K-1) \frac{n}{K} q_0 + O_p(p_0 \frac{n\sqrt{\log n}}{K\sqrt{d}} + \sqrt{p_0 q_0} \frac{n\sqrt{\log n}}{\sqrt{d}} \\
& + K q_0 \frac{n\sqrt{\log n}}{\sqrt{d}}) \big).
\end{aligned}
\tag{43}
$$

To sum up, we arrive at

$$
\begin{aligned}
& \cos\theta(H_{i_1}^{(1)}, H_{i_2}^{(1)}) \\
= \ & \frac{\langle H_{i_1}^{(1)}, H_{i_2}^{(1)} \rangle}{\sqrt{\langle H_{i_1}^{(1)}, H_{i_1}^{(1)} \rangle \cdot \langle H_{i_2}^{(1)}, H_{i_2}^{(1)} \rangle}} \\
= \ & \frac{\frac{n}{k} p_0^2 + (K-1) \frac{n}{K} q_0^2}{\frac{n}{k} p_0 + (K-1) \frac{n}{K} q_0 + O_p(p_0 \frac{n\sqrt{\log n}}{K\sqrt{d}} + \sqrt{p_0 q_0} \frac{n\sqrt{\log n}}{\sqrt{d}} + K q_0 \frac{n\sqrt{\log n}}{\sqrt{d}})} \\
= \ & \underbrace{\frac{\frac{n}{k} p_0^2 + (K-1) \frac{n}{K} q_0^2}{\frac{n}{k} p_0 + (K-1) \frac{n}{K} q_0}}_{\text{cut}_1(1)} + o_p(1)
\end{aligned}
\tag{44}
$$

for $i_1, i_2$ from the same group, when $d \gg \log n$. Similarly, we have

$$
\begin{aligned}
& \cos\theta(H_{i_1}^{(1)}, H_{i_2}^{(2)}) \\
= \ & \frac{\langle H_{i_1}^{(1)}, H_{i_2}^{(1)} \rangle}{\sqrt{\langle H_{i_1}^{(1)}, H_{i_1}^{(1)} \rangle \cdot \langle H_{i_2}^{(1)}, H_{i_2}^{(1)} \rangle}} \\
= \ & \frac{\frac{n}{k} p_0^2 + (K-1) \frac{n}{K} q_0^2 + O_p(p_0 \frac{n\sqrt{\log n}}{K\sqrt{d}} + \sqrt{p_0 q_0} \frac{n\sqrt{\log n}}{\sqrt{d}} + K q_0 \frac{n\sqrt{\log n}}{\sqrt{d}})}{\frac{n}{k} p_0 + (K-1) \frac{n}{K} q_0} \\
= \ & \underbrace{\frac{2 \frac{n}{k} p_0 q_0 + (K-2) \frac{n}{K} q_0^2}{\frac{n}{k} p_0 + (K-1) \frac{n}{K} q_0}}_{\text{cut}_2(1)} + o_p(1)
\end{aligned}
\tag{45}
$$

for $i_1, i_2$ from different groups, when $d \gg \log n / p_0^2$.

Therefore, for any fixed $L \geq$ and any fixed cutoff $\tau \geq \text{cut}_1(L)$, then SERA will predict at least $pK\frac{n}{K} \cdot (\frac{n}{K} - 1)/2 + qK(K-1)/2 + \frac{n}{K} \cdot \frac{n}{K}$ truly connected pairs as dis-connected. In other words,

we have the false negative rate is at least $p/(2k)+q/2$. If the cutoff $\tau$ is between $\text{cut}_2(L)$ and $\text{cut}_1(L)$, then SERA will predict at least $(1-p)K\frac{n}{K}\cdot(\frac{n}{K}-1)/2$ truly dis-connected pairs as connected and predict at least $qK(K-1)/2+\frac{n}{K}\cdot\frac{n}{K}$ truly connected pairs as dis-connected. That is, false positive rate is at least $(1-p)/(2k)$ and false negative rate is at least $(1-q)/2$. If the cutoff $\tau$ is less than $\text{cut}_2(L)$, then SERA will predict at least $(1-p)K\frac{n}{K}\cdot(\frac{n}{K}-1)/2+(1-q)K(K-1)/2+\frac{n}{K}\cdot\frac{n}{K}$ truly connected pairs as dis-connected. That is, false positive rate is at least $(1-p)/(2k)+(1-q)/2$. This completes the proof.

## C.3   Proof of theorem 6.1

As mentioned in section 6, NAG actually protects against a much stronger class of adversaries. Specifically, let $\mathbf{H}=\{H^{(l)}\}_{0\le l\le L}$ denote all the intermediate representations produced by the underlying GNN with weights $\mathbf{W}=\{W_l\}_{l\in[L]}$. The following theorem is a stronger version of theorem 6.1:

**Theorem C.8.** *Assume the adversary $\mathcal{A}$ has access to $\mathbf{H}$ and $\mathbf{W}$, and outputs an estimate of graph adjacencies $\widehat{A}=\mathcal{A}(\mathbf{H},\mathbf{W})$. Then for any graph $G$ and any such adversary $\mathcal{A}$, we have the lower bound:*

$$\min_{u\in V,v\in V}\left[\mathbb{P}\left(\widehat{A}_{uv}=1|A_{uv}=0\right)+\mathbb{P}\left(\widehat{A}_{uv}=0|A_{uv}=1\right)\right]\ge 1-\sqrt{1-\exp\left(-C\frac{\sum_{l\in[L]}\|W_l\|_{op}^2}{\sigma^2}\right)}. \quad (46)$$

*Here the constant $C$ depends on the AGG mechanism of the GNN. In particular, for some standard GNN architectures we have: $C_{GCN}=C_{MEAN\text{-}SAGE}=C_{GIN}=1$ and $C_{GAT}=C_{MAX\text{-}SAGE}=4$.*

The theorem is a consequence of the following lemma:

**Lemma C.9.** *Fix an arbitrary node pair $(u,v)$. Let $\mathbf{H}_1$ and $\mathbf{H}_0$ be the collection of node representations generated under $A_{uv}=1$ and $A_{uv}=0$, respectively. It follows that the Kullback-Leibler divergence between $\mathbf{H}_1$ and $\mathbf{H}_0$ is bounded:*

$$D_{KL}\left(\mathbf{H}_1\parallel\mathbf{H}_0\right)\le C\frac{\sum_{l\in[L]}\|W_l\|_{op}^2}{\sigma^2}. \quad (47)$$

*Here the constant $C=1$ for (SAGE-meanpool), (GIN) and (GCN); and $C=4$ for (SAGE-maxpool) and (GAT).*

*Proof of lemma C.9.* The proof is essentially a proof of Rényi differential privacy similar to that in [36]. First we fix a single $l$-th layer of GNN defined in (8). We rewrite (8) as:

$$H_v^{(l)}=\text{Act}\left(\text{AGG}\left(\frac{W_lH_u^{(l-1)}}{\left\|H_u^{(l-1)}\right\|_2},u\in N(v)\cup\{v\}\right)+\epsilon\right):=\text{Act}\left(\tilde{H}_v^{(l-1)}+\epsilon\right) \quad (48)$$

Let the corresponding representation matrix be $H_1^{(l)}$ for $A_{uv}=1$ and $H_0^{(l)}$ for $A_{uv}=0$ for any $l\in[L]$. Further denote $\widetilde{H}_a^l=\{\tilde{H}_{v,a}^{(l)}\}_{v\in V}$ as the intermediate representation defined as in (48) with $A_{uv}=a,a\in\{0,1\}$. Then by standard results on Rényi divergence [24], we have

$$D_{KL}\left(H_1^l\parallel H_0^l\right)=\frac{\left\|\widetilde{H}_1^{(l)}-\widetilde{H}_0^{(l)}\right\|_2^2}{2\sigma^2} \quad (49)$$

For some input $H^{l-1}$. It follows that given all the other edges, the only terms that contributes to $\left\|\widetilde{H}_1^{(l)}-\widetilde{H}_0^{(l)}\right\|_2^2$ are $\left\|\tilde{H}_{v,1}^{(l)}-\tilde{H}_{v,0}^{(l)}\right\|_2^2$ and $\left\|\tilde{H}_{u,1}^{(l)}-\tilde{H}_{u,0}^{(l)}\right\|_2^2$. Next we give the derivation of various GNN architectures:

**The case of** (SAGE-meanpool) We let $d_v$ to be the degree of $v$ assuming $A_{uv} = 1$. Further let $g_v^{(l)} = \frac{W_l h_u^{(l-1)}}{\left\| h_u^{(l-1)} \right\|_2}$ We have:

$$\left\| \tilde{H}_{u,1}^{(l)} - \tilde{H}_{u,0}^{(l)} \right\|_2 = \left\| \frac{1}{d_v + 1} \left( g_v^{(l-1)} - \frac{1}{d_v} \sum_{u \in \overline{N}(v) \backslash \{v\}} g_u^{(l-1)} \right) \right\|_2 \tag{50}$$

$$\leq \frac{1}{2} \left( \left\| g_v^{(l-1)} \right\|_2 + \frac{1}{d_v} \sum_{u \in \overline{N}(v) \backslash \{v\}} \left\| g_u^{(l-1)} \right\|_2 \right) \tag{51}$$

$$\leq \frac{1}{2} \left( \|W_l\|_{\text{op}} + \frac{1}{d_v} \sum_{u \in \overline{N}(v) \backslash \{v\}} \|W_l\|_{\text{op}} \right) \tag{52}$$

$$= \|W_l\|_{\text{op}} \tag{53}$$

Analogously we have $\left\| \tilde{H}_{u,1}^{(l)} - \tilde{H}_{u,0}^{(l)} \right\|_2^2 \leq \|W_l\|_{\text{op}}^2$ and thus $D_{\text{KL}} \left( H_1^{(l)} \parallel H_0^{(l)} \right) \leq \frac{\|W_l\|_{\text{op}}^2}{\sigma^2}$. The result follows from adaptive composition as in [24, Proposition 1].

**The case of** (GIN) This follows by combining the preceding argument with [36, Proposition 1].

**The case of** (GCN) This follows by combining the preceding argument with [36, Proposition 2].

**The case of** (SAGE-maxpool) The result follows from the following fact that $\left\| \max_{u \in \overline{N}(v)} g_u - \max_{u \in \overline{N} \backslash \{v\}} g_u \right\|_2$ attains its maximum when $g_v = -g_u, \forall u \in \overline{N} \backslash \{v\}$ since all the $g_u$s are unit vectors.

$\square$

*Proof of theorem 6.1.* We view the reconstruction problem regarding $A_{uv}$ as a binary hypothesis testing problem

$$H_0 : A_{uv} = 0 \qquad \text{v.s.} \qquad H_1 : A_{uv} = 1. \tag{54}$$

Then according to hypothesis testing theory [21], we have

$$\inf_{\widehat{A}} \left[ \mathbb{P} \left( \widehat{A}_{uv} = 1 | A_{uv} = 0 \right) + \mathbb{P} \left( \widehat{A}_{uv} = 0 | A_{uv} = 1 \right) \right] \geq 1 - d_{\text{TV}} \left( \mathbf{H}_1, \mathbf{H}_0 \right), \tag{55}$$

where we use $d_{\text{TV}} \left( \mathbf{H}_1, \mathbf{H}_0 \right)$ to denote the total variation distance of distributions induced by $\mathbf{H}_1$ and $\mathbf{H}_0$ respectively. By the Bretagnolle–Huber bound [4, Theorem 1], we have

$$d_{\text{TV}} \left( \mathbf{H}_1, \mathbf{H}_0 \right) \leq \sqrt{1 - \exp \left( -D_{\text{KL}} \left( \mathbf{H}_1 \parallel \mathbf{H}_0 \right) \right)} \tag{56}$$

The result then follows by combining (55), (56) and lemma C.9. $\square$

# D Further experiments

## D.1 Synthetic experiments

**Erdős–Rényi experiments with random GNN weights** The experimental setup in this section is basically the same as that in section 7.1, except that the model weights are generated by the following process: For an $L$-layer Linear GNN, we generate the weight matrix as:

$$W = W_1 \times \cdots \times W_L. \tag{57}$$

Here each $W_l, 1 \leq l \leq 10$ is a random matrix generated using the initialization method proposed in [16]. The evaluations are shown in figure 5.

The results exhibit a similar pattern to figure 4 where the weight matrix is set to identity. However, the attacking performance differs between the two scenarios: When the matrix $W$ is poorly conditioned (a consequence of the construction (57)), the attacking performance degrades especially when the feature dimension $d$ is not sufficiently large.

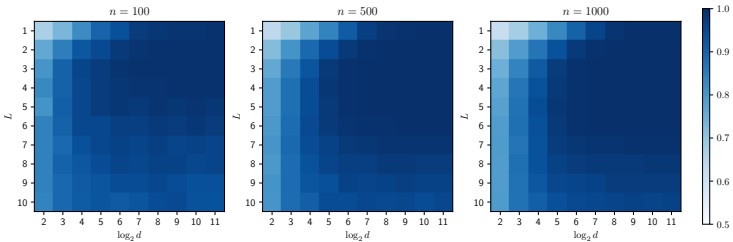

(a) Measured in AUROC metric, darker color implies higher attacking performance

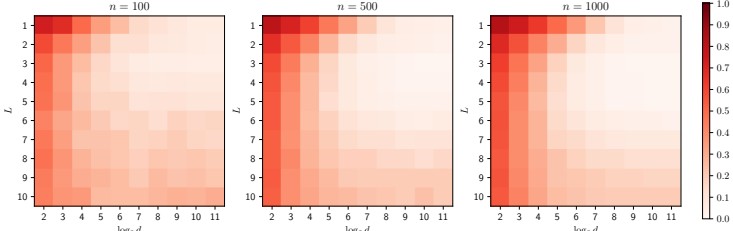

(b) Measured in ERR metric, lighter color implies higher attacking performance

Figure 4: Attacking efficacy of SERA over sparse Erdős–Rényi graphs, with each grid's value indicating SERA's performance measured in either AUROC (first row) or ERR (second row) metric.



(a) Measured in AUROC metric, darker color implies higher attacking performance

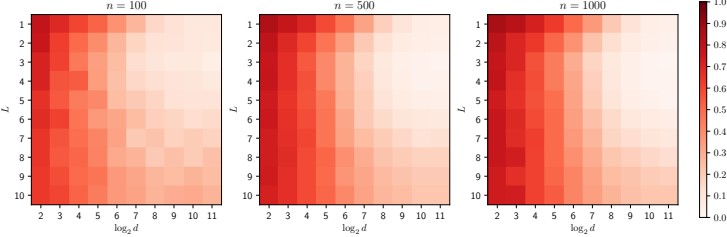

(b) Measured in ERR metric, lighter color implies higher attacking performance

Figure 5: Attacking efficacy of SERA over sparse Erdős–Rényi graphs, with each grid's value indicating SERA's performance measured in either AUROC (first row) or ERR (second row) metric.

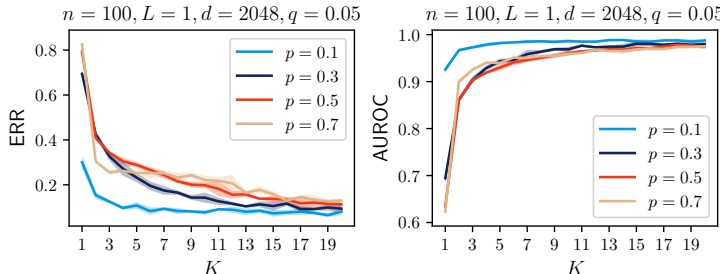

Figure 6: Performance of SERA on SBM with varying $K$ and $p$. All plots are based on 5 independent trials with shades indicating one standard deviation.

Table 2: Summary of dataset characteristics

|            | Squirrel | Chameleon | Actor | Cora  | Citeseer | Pubmed | Products  | Reddit    |
|------------|----------|-----------|-------|-------|----------|--------|-----------|-----------|
| # nodes    | 5201     | 2277      | 7600  | 2708  | 3327     | 19717  | 1569960   | 232965    |
| # edges    | 36101    | 217073    | 30019 | 10556 | 9104     | 88648  | 264339468 | 114615892 |
| # features | 2089     | 2325      | 932   | 1433  | 3703     | 500    | 200       | 602       |
| # classes  | 5        | 5         | 5     | 7     | 6        | 3      | 107       | 41        |

**Impact of SBM structure** To investigate the impact of SBM structure on the performance of SERA, we fix the GNN architecture at $L = 1$ and evaluate on a graph with 100 nodes and node feature dimension $d = 2048$. Note that we choose a relatively large node feature dimension to ensure that the assumption listed in theorem 5.1 is approximately met. We vary the SBM within-group probability according to $p \in \{0.1, 0.3, 0.5, 0.7\}$ and the number of groups according to $1 \leq K \leq 20$. The results, plotted in figure 6, suggest that in general, the attacking performance is positively correlated with the number of groups $K$ since more groups yield stronger sparsity according to the SBM generation law. This phenomenon is also in accordance with theorem 5.1.

### D.2 A complete report of attack performance on real-world datasets

**Dataset characteristics** We report a brief summary of datasets used in table 2.

**Training configurations** Across all experiments we use a hidden dimension of $d = 128$ with number of GNN layers adjusted to $L \in \{2, 5\}$. We use Adam optimizer with learning rate 0.001 and train for 1000 epochs on the 6 relatively small datasets, and train for 5 epochs on Amazon-Products and Reddit datasets, using a stochastic training strategy that samples up to 20 neighbors per node in each layer.

**Results** We present a full list of results in table 3. A comprehensive tabulation of outcomes is furnished in Table 3. Consistent with the findings explicated in section 7, SERA demonstrates proficient reconstruction capabilities irrespective of graph properties such as homophily. Additionally, the reconstruction potency is resilient across a spectrum of GNN architectures. These results imply that prevalent GNN architectures are likely to engender models that are vulnerable to exploitation by SERA adversaries.

### D.3 A complete report of privacy-utility assessments on Planetoid datasets

#### D.3.1 Training configurations and attacking pipeline

**Network design** For node $v$ with label $y_v$, the prediction is defined as

$$\hat{y}_v = \arg \max_{c \in [C]} \mathsf{dec} \left( \mathsf{enc} \left( G, \mathbf{W} \right) [v] \right) [c], \tag{58}$$

where we use $[\cdot]$ to denote the operation of vector index. Here the encoder $\mathsf{enc}$ is designed via stacking $L$ noisy GNN layers (in the sense of NAG) with aggregation mechanism $\mathsf{AGG} \in$

Table 3: Performances of SERA on eight datasets measured by AUROC metric (%). For each setup, the results (in the form of $\text{mean}_{\pm\text{std}}$) are obtained via 5 random trials.

| | Squirrel | Chameleon | Actor | Cora | Citeseer | Pubmed | Products | Reddit |
|---|---|---|---|---|---|---|---|---|
| $\mathcal{H}_{\text{label}}$ | 0.22 | 0.23 | 0.22 | 0.81 | 0.74 | 0.8 | 0.09 | 0.76 |
| $\mathcal{H}_{\text{feature}}$ | 0.01 | 0.01 | 0.16 | 0.17 | 0.19 | 0.27 | 0.01 | 0.12 |
| $\widehat{A}^{\text{FS}}$ | 46.2 | 55.2 | 44.7 | 80.3 | 87.4 | 87.6 | 52.0 | 95.9 |
| Victim model | $\widehat{A}^{\text{SERA}}$, non-trained | | | | | | | |
| LIN($L=2$) | $72.8_{\pm0.0}$ | $76.1_{\pm0.2}$ | $73.1_{\pm0.1}$ | $93.1_{\pm0.4}$ | $92.5_{\pm0.9}$ | $93.9_{\pm1.2}$ | $97.2_{\pm0.3}$ | $96.4_{\pm0.1}$ |
| LIN($L=5$) | $72.6_{\pm0.0}$ | $76.0_{\pm0.3}$ | $73.0_{\pm0.2}$ | $95.9_{\pm0.6}$ | $93.8_{\pm0.4}$ | $96.0_{\pm0.6}$ | $99.2_{\pm0.1}$ | $95.4_{\pm0.1}$ |
| GCN($L=2$) | $87.3_{\pm0.3}$ | $87.9_{\pm0.4}$ | $87.1_{\pm0.6}$ | $99.8_{\pm0.1}$ | $99.9_{\pm0.0}$ | $99.7_{\pm0.0}$ | $99.6_{\pm0.0}$ | $97.3_{\pm0.1}$ |
| GCN($L=5$) | $82.1_{\pm0.3}$ | $84.3_{\pm0.9}$ | $84.1_{\pm0.9}$ | $99.4_{\pm0.2}$ | $99.9_{\pm0.0}$ | $99.5_{\pm0.1}$ | $99.2_{\pm0.1}$ | $96.1_{\pm0.2}$ |
| GAT($L=2$) | $86.2_{\pm0.5}$ | $87.9_{\pm0.5}$ | $86.7_{\pm0.6}$ | $99.8_{\pm0.1}$ | $99.9_{\pm0.0}$ | $99.7_{\pm0.0}$ | $79.8_{\pm7.7}$ | $88.5_{\pm8.6}$ |
| GAT($L=5$) | $82.0_{\pm0.0}$ | $84.3_{\pm0.4}$ | $82.9_{\pm0.7}$ | $99.5_{\pm0.1}$ | $99.9_{\pm0.0}$ | $99.4_{\pm0.0}$ | $91.3_{\pm5.9}$ | $93.7_{\pm1.7}$ |
| GIN($L=2$) | $72.8_{\pm0.0}$ | $76.1_{\pm0.2}$ | $73.5_{\pm0.1}$ | $92.1_{\pm0.8}$ | $91.3_{\pm0.9}$ | $96.6_{\pm0.6}$ | $98.8_{\pm0.1}$ | $96.4_{\pm0.1}$ |
| GIN($L=5$) | $72.2_{\pm0.0}$ | $75.9_{\pm0.4}$ | $74.4_{\pm0.0}$ | $93.0_{\pm0.3}$ | $89.9_{\pm0.7}$ | $94.2_{\pm0.5}$ | $97.6_{\pm0.1}$ | $89.2_{\pm0.6}$ |
| SAGE($L=2$) | $72.6_{\pm0.1}$ | $75.5_{\pm0.1}$ | $74.1_{\pm0.2}$ | $87.6_{\pm0.3}$ | $87.0_{\pm0.9}$ | $93.7_{\pm0.9}$ | $99.2_{\pm0.0}$ | $94.2_{\pm0.5}$ |
| SAGE($L=5$) | $71.4_{\pm0.2}$ | $74.2_{\pm0.1}$ | $72.4_{\pm0.6}$ | $88.4_{\pm1.2}$ | $85.6_{\pm0.3}$ | $88.4_{\pm1.0}$ | $97.6_{\pm0.1}$ | $93.4_{\pm0.6}$ |
| Victim model | $\widehat{A}^{\text{SERA}}$, trained | | | | | | | |
| LIN($L=2$) | $74.6_{\pm0.0}$ | $75.0_{\pm0.3}$ | $59.9_{\pm0.7}$ | $94.6_{\pm0.1}$ | $93.7_{\pm0.1}$ | $89.0_{\pm0.1}$ | $91.6_{\pm0.2}$ | $94.7_{\pm0.1}$ |
| LIN($L=5$) | $74.1_{\pm0.3}$ | $76.9_{\pm0.2}$ | $61.6_{\pm0.7}$ | $94.8_{\pm0.3}$ | $93.3_{\pm0.1}$ | $88.4_{\pm0.9}$ | $98.6_{\pm0.1}$ | $92.3_{\pm0.2}$ |
| GCN($L=2$) | $79.4_{\pm0.4}$ | $82.3_{\pm0.3}$ | $78.5_{\pm0.8}$ | $97.8_{\pm0.1}$ | $99.0_{\pm0.0}$ | $89.2_{\pm0.3}$ | $94.5_{\pm0.1}$ | $95.1_{\pm0.1}$ |
| GCN($L=5$) | $77.4_{\pm0.6}$ | $80.6_{\pm0.8}$ | $78.4_{\pm0.6}$ | $97.4_{\pm0.3}$ | $98.7_{\pm0.2}$ | $92.6_{\pm0.4}$ | $98.4_{\pm0.1}$ | $95.0_{\pm0.1}$ |
| GAT($L=2$) | $73.3_{\pm0.9}$ | $74.3_{\pm1.0}$ | $66.4_{\pm0.6}$ | $95.2_{\pm0.2}$ | $96.2_{\pm0.1}$ | $89.3_{\pm0.4}$ | $98.2_{\pm0.6}$ | $95.7_{\pm0.2}$ |
| GAT($L=5$) | $79.2_{\pm0.6}$ | $77.2_{\pm1.3}$ | $75.0_{\pm2.1}$ | $95.6_{\pm0.2}$ | $95.9_{\pm0.6}$ | $91.6_{\pm0.8}$ | $84.5_{\pm10.1}$ | $88.5_{\pm4.6}$ |
| GIN($L=2$ | $73.3_{\pm0.5}$ | $77.2_{\pm0.5}$ | $71.5_{\pm0.3}$ | $94.2_{\pm0.3}$ | $95.3_{\pm0.1}$ | $91.0_{\pm0.1}$ | $99.0_{\pm0.1}$ | $95.5_{\pm0.4}$ |
| GIN($L=5$) | $73.2_{\pm0.3}$ | $77.0_{\pm0.6}$ | $74.1_{\pm0.1}$ | $99.8_{\pm0.1}$ | $94.1_{\pm0.3}$ | $92.7_{\pm1.5}$ | $97.7_{\pm0.1}$ | $92.4_{\pm0.9}$ |
| SAGE($L=2$) | $71.6_{\pm0.1}$ | $71.7_{\pm1.3}$ | $67.8_{\pm0.7}$ | $91.1_{\pm0.2}$ | $93.3_{\pm0.1}$ | $87.0_{\pm0.2}$ | $96.2_{\pm0.4}$ | $95.4_{\pm0.1}$ |
| SAGE($L=5$) | $71.6_{\pm0.1}$ | $74.5_{\pm0.2}$ | $71.2_{\pm0.6}$ | $93.6_{\pm0.5}$ | $93.4_{\pm0.8}$ | $85.5_{\pm1.0}$ | $96.6_{\pm0.8}$ | $87.5_{\pm1.7}$ |

$\{\textsf{MEAN}, \textsf{SUM}, \textsf{GCN}, \textsf{ATTENTION}, \textsf{MAX}\}$ as defined above. Note that the encoder maps input node features into node representations of dimension $d$, which might be larger than the number of classes $C$. The decoder dec is a linear map that maps node representations to predictions.

**Attacking paradigm** The attacking procedure of SERA will be based on the node representations produced by the GNN encoder enc under a dimension of $d$. The attack is conducted over the node representations corresponding to the test subset, i.e., the victim subgraph is the subgraph induced by the test nodes.

**Training configurations** Across all the experiments, we fix the GNN model to be of depth 2 and use full-batch training for 1000 steps(epochs) using the Adam optimizer with a learning rate of 0.001.

### D.3.2 Unconstrained scheme

We plot the full experimental results under the unconstrained scheme for the Cora, Citeseer and Pubmed datasets in figure 8, figure 9 and figure 10, respectively, where we evaluate the performance of SERA under both ERR and AUROC metrics. The result is consistent with those findings listed in section 7.3.

### D.3.3 Constrained scheme

We plot the full experimental results under the constrained scheme for the Cora, Citeseer and Pubmed datasets in figure 11, figure 12 and figure 13, respectively, where we evaluate the performance of SERA under both ERR and AUROC metrics. The result is consistent with those findings listed in section 7.3.

**Impact of different AGG mechanisms** According to figure 11, 8, 12, 9, 13, 10, the previously discovered phenomenons are present for all the 5 aggregation types. Nevertheless, the degree to which these phenomena exhibit varies with the specific type of aggregation employed. Notably, the

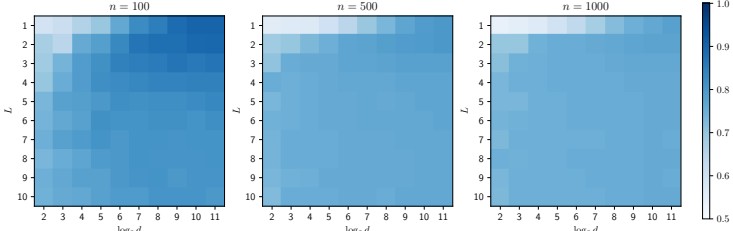

(a) Measured in AUROC metric, darker color implies higher attacking performance

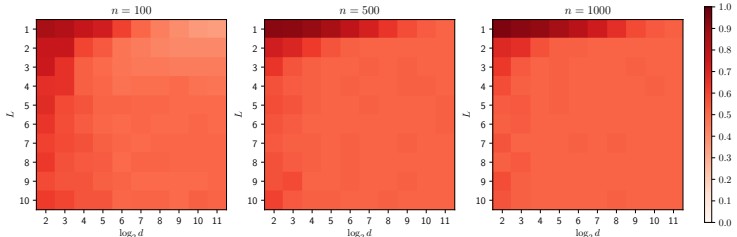

(b) Measured in ERR metric, lighter color implies higher attacking performance

Figure 7: Attacking efficacy of SERA over dense SBM graphs, with each grid's value indicating SERA's performance measured in either AUROC (first row) or ERR (second row) metric.

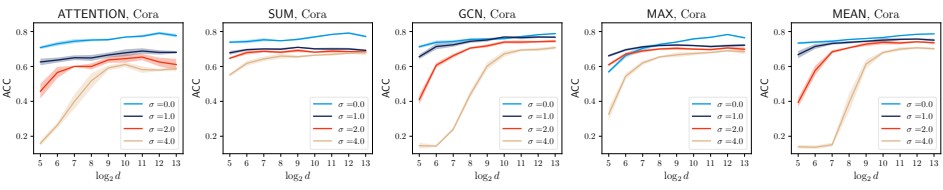

(a) GNN model performance over Cora dataset under 5 different aggregation types.

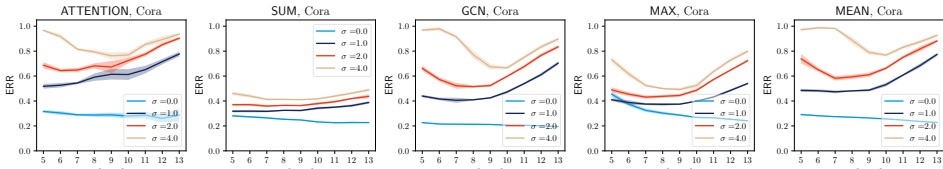

(b) Attacking performance of SERA over Cora dataset (measured by ERR) under 5 different aggregation types.

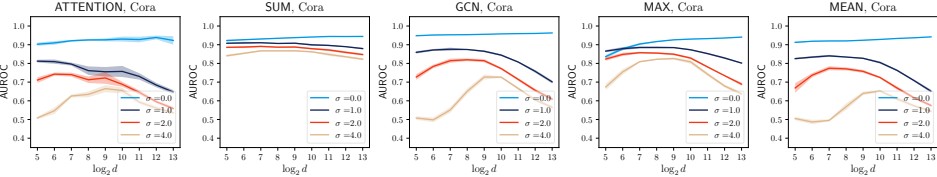

(c) Attacking performance of SERA over Cora dataset (measured by AUROC) under 5 different aggregation types.

Figure 8: Privacy-utility trade-off on Cora dataset using the unconstrained training scheme. The horizontal axes measure feature dimension $d$ in $\log_2$ scale and the vertical axes stands for performance measures All plots are based on 5 independent trials with shades indicating one standard deviation.

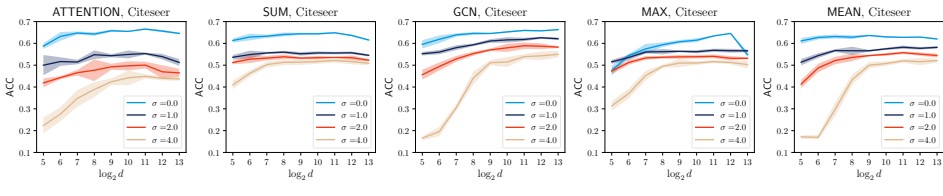

(a) GNN model performance over Citeseer dataset under 5 different aggregation types.

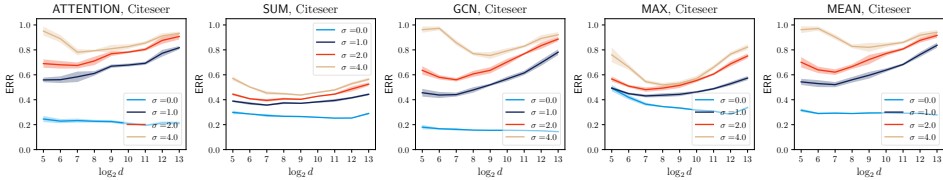

(b) Attacking performance of SERA over Citeseer dataset (measured by ERR) under 5 different aggregation types.

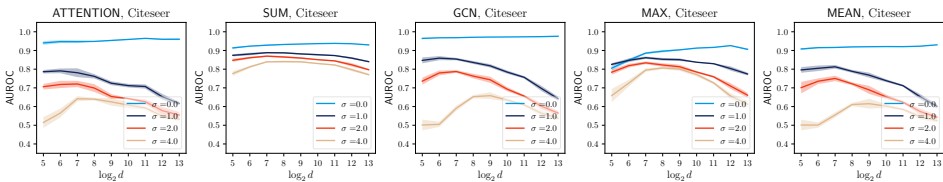

(c) Attacking performance of SERA over Citeseer dataset (measured by AUROC) under 5 different aggregation types.

Figure 9: Privacy-utility trade-off on Citeseer dataset using the unconstrained training scheme. The horizontal axes measure feature dimension $d$ in $\log_2$ scale and the vertical axes stands for performance measures All plots are based on 5 independent trials with shades indicating one standard deviation.

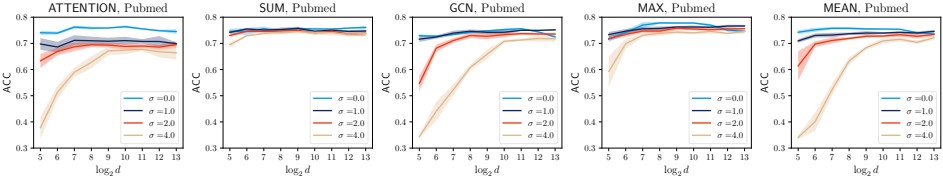

(a) GNN model performance over Pubmed dataset under 5 different aggregation types.

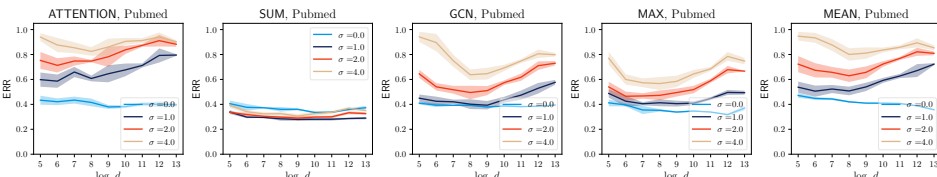

(b) Attacking performance of SERA over Pubmed dataset (measured by ERR) under 5 different aggregation types.

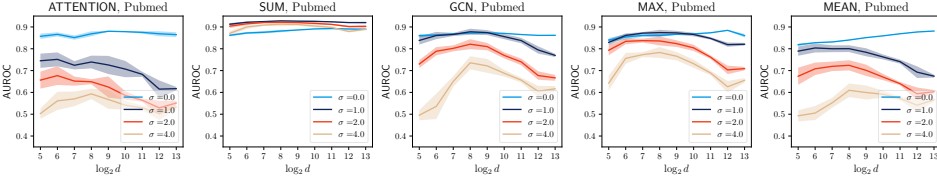

(c) Attacking performance of SERA over Pubmed dataset (measured by AUROC) under 5 different aggregation types.

Figure 10: Privacy-utility trade-off on Pubmed dataset using the unconstrained training scheme. The horizontal axes measure feature dimension $d$ in $\log_2$ scale and the vertical axes stands for performance measures All plots are based on 5 independent trials with shades indicating one standard deviation.

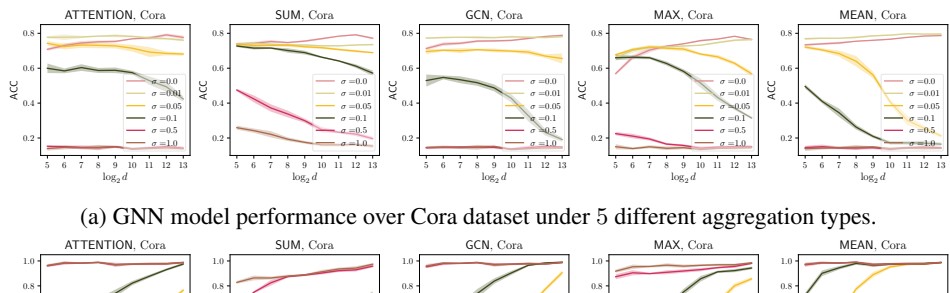

(a) GNN model performance over Cora dataset under 5 different aggregation types.

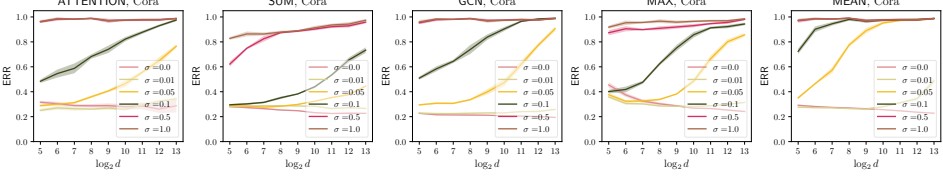

(b) Attacking performance of SERA over Cora dataset (measured by ERR) under 5 different aggregation types.

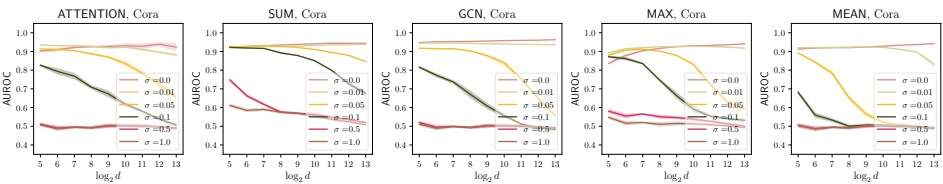

(c) Attacking performance of SERA over Cora dataset (measured by AUROC) under 5 different aggregation types.

Figure 11: Privacy-utility trade-off on Cora dataset using the constrained training scheme. The horizontal axes measure feature dimension $d$ in $\log_2$ scale and the vertical axes stands for performance measures All plots are based on 5 independent trials with shades indicating one standard deviation.

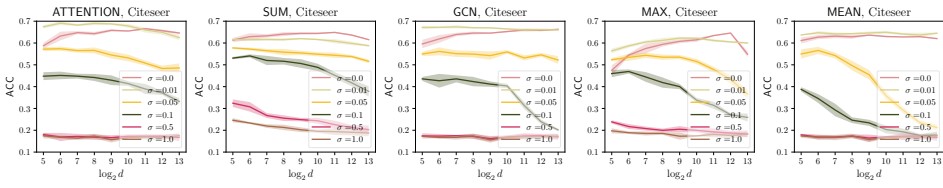

(a) GNN model performance over Citeseer dataset under 5 different aggregation types.

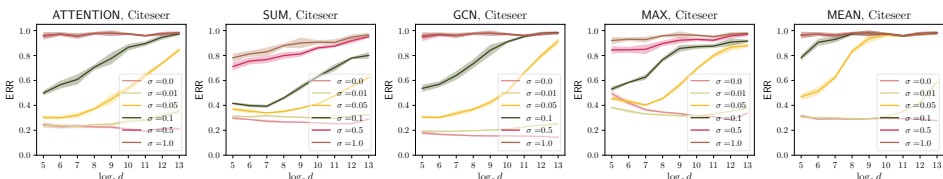

(b) Attacking performance of SERA over Citeseer dataset (measured by ERR) under 5 different aggregation types.

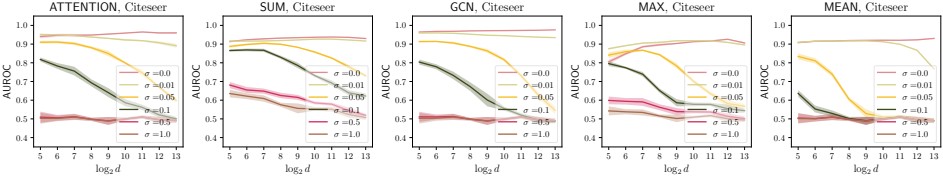

(c) Attacking performance of SERA over Citeseer dataset (measured by AUROC) under 5 different aggregation types.

Figure 12: Privacy-utility trade-off on Citeseer dataset using the constrained training scheme. The horizontal axes measure feature dimension $d$ in $\log_2$ scale and the vertical axes stands for performance measures All plots are based on 5 independent trials with shades indicating one standard deviation.

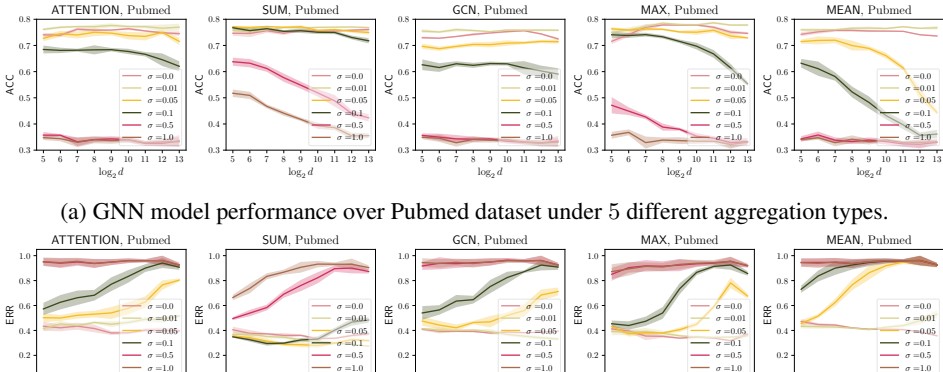

(a) GNN model performance over Pubmed dataset under 5 different aggregation types.

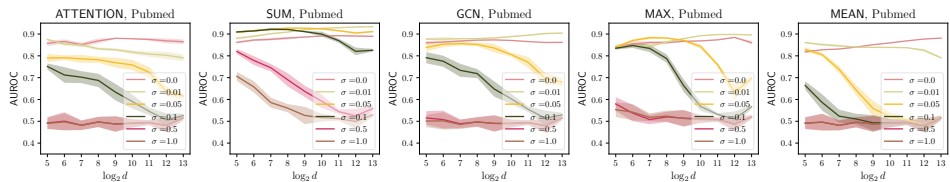

(b) Attacking performance of SERA over Pubmed dataset (measured by ERR) under 5 different aggregation types.

(c) Attacking performance of SERA over Pubmed dataset (measured by AUROC) under 5 different aggregation types.

Figure 13: Privacy-utility trade-off on Pubmed dataset using the constrained training scheme. The horizontal axes measure feature dimension $d$ in $\log_2$ scale and the vertical axes stands for performance measures All plots are based on 5 independent trials with shades indicating one standard deviation.

behaviors of ATTENTION, MEAN, and GCN pooling display similarities attributable to their shared mechanism in (weighted) average aggregation. Conversely, the efficacy of the SERA against Noisy Aggregation (NAG) when SUM and MAX pooling are utilized appears less susceptible to changes in $d$.

## D.4 Spectrum study of GNN solutions obtained under the unconstrained scheme

**A closer look at GNN solutions obtained via NAG in the unconstrained scheme** As SERA is just one form of attack mechanism under a weak adversary, protecting against SERA does not necessarily imply strict notions of privacy. Motivated by theorem 6.1, we conduct a spectrum study regarding the GNN solutions obtained via NAG in the unconstrained scheme. Specifically, we plot the operator norm of the weight matrices corresponding to the GNN layers across all scenarios and report them in the last column in figure 14 in appendix D.4. The results exhibit a rapidly growing trend of weights' operator norms regarding the increase of both feature dimension $d$ and noise level $\sigma$. For GNN models trained using noisy aggregation under large $d$s, the corresponding bounds according to (8) become vacuous, i.e., practically zero. Additionally, these solutions may exhibit diminished robustness, as the corresponding Lipschitz constants are likely to be inadequately regulated [40]. To conclude, we have found successful empirical defenses against SERA without satisfying strict notions of privacy, suggesting that **SERA has limitations as a tool for auditing private GRL training procedures**.

## D.5 Privacy-utility trade-off comparisons: NAG vs EdgeRR

In this section we provide preliminary comparisons between NAG and EdgeRR regarding their privacy-utility trade-offs. In particular, for a graph with $n$ nodes, the EdgeRR with budget $\varepsilon$ is implemented as a graph-level transform that perturbs the adjacency matrix $\mathsf{EdgeRR}(A) \in \mathbb{R}^{n \times n}$

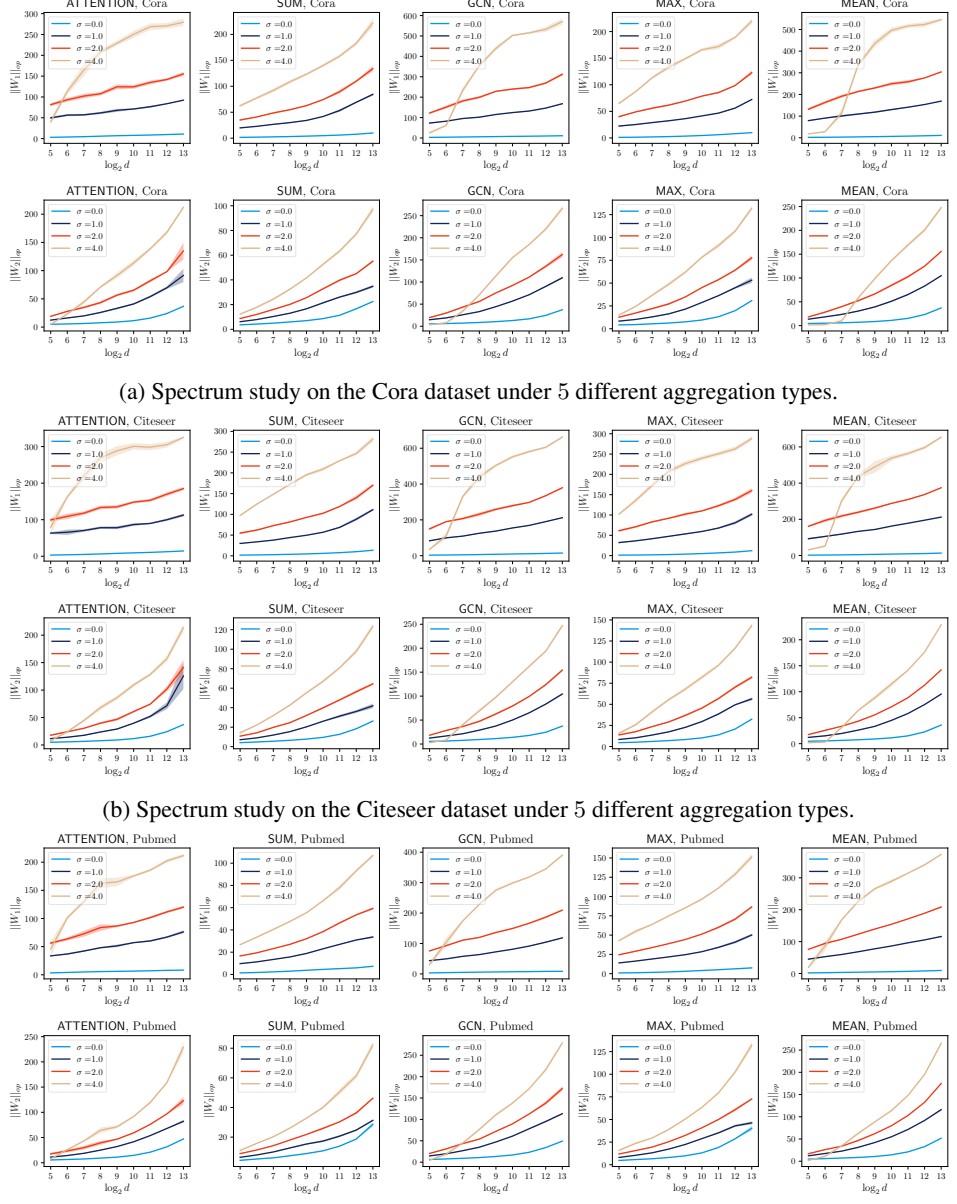

(a) Spectrum study on the Cora dataset under 5 different aggregation types.

(b) Spectrum study on the Citeseer dataset under 5 different aggregation types.

(c) Spectrum study on the Pubmed dataset under 5 different aggregation types.

Figure 14: Spectrum study on the Planetoid datasets under the unconstrained training scheme. The horizontal axes measure feature dimension $d$ in $\log_2$ scale and the vertical axes measures the operator norm of the projection weights of the GNN. All plots are based on 5 independent trials with shades indicating one standard deviation.

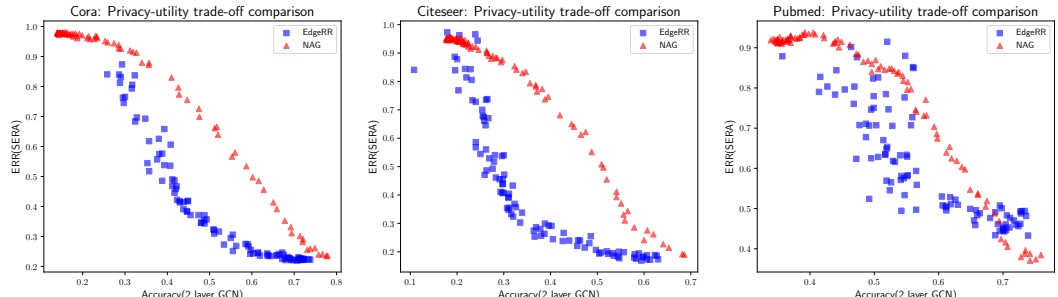

Figure 15: Comparison of privacy-utility trade-off between NAG and EdgeRR

with each of its entries defined as:

$$\text{EdgeRR}(A)_{u,v} = \begin{cases} A_{u,v} & \text{With probability } \frac{e^\varepsilon}{1+e^\varepsilon} \\ 1 - A_{u,v} & \text{Otherwise} \end{cases} \tag{59}$$

It then follows from the theory of local differential privacy [19] that for any $(u, v)$ the perturbed entry is a $\varepsilon$-LDP view of the underlying true adjacency. Combining the property of max-divergence along with the proof techique in section C.3 we have the following performance lower bound of any adversary $\mathcal{A}$:

$$\inf_{\mathcal{A}} \min_{u \in V, v \in V} \left[ \mathbb{P}\left( \widehat{A}_{uv} = 1 | A_{uv} = 0 \right) + \mathbb{P}\left( \widehat{A}_{uv} = 0 | A_{uv} = 1 \right) \right] \geq 1 - \sqrt{1 - e^{-\varepsilon}}. \tag{60}$$

While EdgeRR has a very strong privacy protection guarantee, it is also criticized for low utility. We provide a comparison using a two-layer GCN as the backbone under the embedding dimension $d = 128$ on the Planetoid datasets. The results, which can be viewed at figure 15, suggest that when considering SERA as the basis for evaluating privacy, NAG achieves a Pareto-dominant privacy-utility trade-off curve relative to EdgeRR.

**Software and hardware infrastructures.** Our framework is built upon PyTorch [28] and PyTorch Geometric [13], which are open-source software released under BSD-style [4] and MIT [5] license, respectively. All datasets used throughout experiments are publicly available. All experiments are done on a single NVIDIA A100 GPU (with 80GB memory).

## E   Discussions and Limitations

### E.1   On the impact of depth $L$ for NAG

As elucidated in theorem 6.1, the privacy assurances provided by NAG are inclined to diminish as the depth parameter $L$ increases, a phenomenon attributable to the compositional nature of (differential) privacy mechanisms [12, 24]. However, this same compositional principle enables NAG to disseminate all intermediate node representations $H^{(1)}, \ldots, H^{(L)}$ while preserving the identical level of privacy as would be the case if only $H^{(L)}$ were released. Consequently, this framework permits the design of superior privacy-preserving GNN architectures by leveraging a blend of $H^l_{l \in [L]}$ through inter-layer aggregation techniques, sometimes termed as residual connections in GRL [39]. Probing the resilience of SERA against such intricate GNN configurations poses considerable challenges and falls outside the purview of this paper. Nonetheless, preliminary evaluations have been executed to discern the ramifications of the depth parameter $L$ on the privacy-utility compromises of NAG, absent residual connections, with privacy quantified via SERA. The inquiries, undertaken using the Planetoid datasets with a fixed noise magnitude of $\sigma = 0.05$ and hidden dimensionality $d = 128$, are visually synthesized in Figure 16. Findings reveal that optimal defensive utility typically transpires at $L = 1$, while greater GNN depths, notably those with $L > 5$, tend to undermine the model's utility. Moreover, the apex of attack performance generally materializes at relatively incipient

---

[4] https://github.com/pytorch/pytorch/blob/master/LICENSE
[5] https://github.com/pyg-team/pytorch_geometric/blob/master/LICENSE

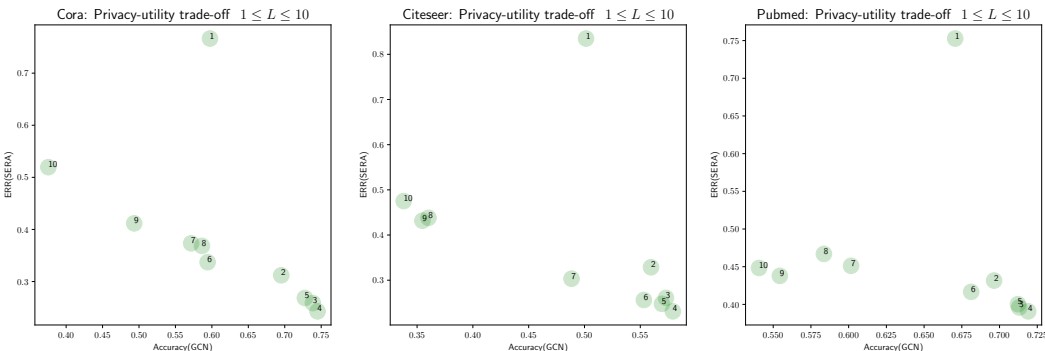

Figure 16: Privacy-utility trade-off on Planetoid regarding different model depths

layers, a confirmation of our theoretical insights set forth in theorem 4.1. Finally, we postulate that incorporating residual connections might proffer an enhanced Pareto frontier for the model. Exploration of this hypothesis is deferred to subsequent research endeavors.

### E.2 Stronger adversary for dense graphs or deep encoders

We have shown the limitations of SERA over dense SBM graphs as well as deep GNN encoders. As our analysis applies to the specific SERA adversary, it is thus of interest to ask whether there exists stronger attacking paradigms that is provably effective against dense graphs or deep GNN encoders. On the flipside, it is also valuable to understand whether the phenomenon of oversmoothing may fundamentally affect the performance of *any* black-box adversary.

### E.3 Extension to more complicated victim GNN models

The theoreical analysis presented in section 4 and section 5 is dedicated to graph neural networks employing mean aggregation without nonlinear activation functions. Prospects exist for augmenting our theoretical framework to encompass alternative aggregation schemes, such as summation [38] and attention-based aggregation [32], conditional upon the satisfaction of specific prerequisites—namely, some lower bound of attention coefficients. A more challenging task lies in the extension of our analysis to graph neural networks (GNNs) that incorporate nonlinear activations between their layers. This inclusion significantly complicates the straightforward application of our non-asymptotic analysis in a cohesive end-to-end manner. Acknowledging the complexity of this endeavor, we left a thorough investigation for future research initiatives.

### E.4 Quantifying the advantage of adversaries with more knowledge

Despite its effectiveness, the knowledge available to SERA is rather limited. Although previous study [17] has shown empirical evidences that equipping the adversary with more capability may results in stronger attacking algorithms, theoretical explication of these enhancements has yet to be articulated. In particular, it is of interest to quantify the amplification of adversarial capacity afforded by scenarios in which the adversary is granted white-box access to the model weights or node features.

