# OpenReview forum: "On provable privacy vulnerabilities of graph representations"
_NeurIPS.cc/2024/Conference — NeurIPS 2024 poster_

### Official Review · Reviewer_w4Le · 2024-07-07

**Soundness:** 3
**Presentation:** 2
**Contribution:** 2
**Rating:** 5
**Confidence:** 3

**Summary:**

This paper investigates the ability of the similarity-based edge reconstruction attack (SERA) on attacking both sparse and dense networks, under various configurations of graph neural network (GNN) structures. The authors demonstrate, through both theoretical analysis and experimental results, that SERA performs well on sparse networks but poorly on dense networks. Additionally, they demonstrate the effectiveness of the noisy aggregation (NAG) technique on the resilience of SERA.

**Strengths:**

This is a initial work to take a step towards a principled theoretical understanding of the effectiveness of SERA over sparse/dense features and different configurations of linear GNN. The theoretical results are presented in a clear and sound way. The proof is rigorous and clearly demonstrated. The explainations following each theorem give good supplements to the theory.

**Weaknesses:**

1. Writing Quality: The writing in this paper needs improvement. The authors have used many unnecessarily complex words and sentences, making the paper harder to read and understand. Additionally, there are numerous typos and grammatical errors that need to be corrected.

2. Theoretical Analysis Limitation: The theoretical analysis focuses on linear GNNs rather than non-linear GNNs. While the authors have shown experimentally that results for linear GNNs can serve as proxies for those of non-linear GNNs, the guarantee of this is unclear to me.

3. Rationale for Studying SERA: In section 2.1, the authors mention that there are attacks more powerful than SERA. This raises the question of why they chose to study SERA in the first place.

4. Accuracy of Statistic in experiment: In Table 1, the authors measure the similarity between feature similarity and edge presence using a statistic $\mathcal{H}$, which only evaluates the similarity of features between nodes with an edge. It does not consider nodes without an edge, leading me to question the accuracy of the statistic to measure similarity and thus the results analysis. In my opinion, it should also capture the dissimilarity of features between nodes without an edge.

**Questions:**

(1) In line 286, it is stated, "Yet the behaviors in small d regimes appear to be less predictable, a phenomenon we hypothesize may be attributable to an inadequate concentration of inner products in instances where the feature dimension is relatively small." Can the authors explain in detail what this means?

---

> ### Author Rebuttal · Authors · 2024-08-05
>
> Thank you for your valuable comments, we will integrate your suggestions into revised versions of our paper. Below we address some specific points:
>
> ## Q1: About the writing style
> Thank you for the advice and we will polish our writing to improve clarity and conciseness.
>
> ## Q2: On the limitations of study linear GNNs
> Please kindly refer to the first point in our global response. Firstly, it is quite challenging to directly analyze the performance of nonlinear GNNs, especially when the underlying GNNs are with more than 1 layers and the dependence on network depth is a critical factor of the study. Secondly, linear GNNs such as SGC constitute an important type of GNN architecture in practice. By characterizing the provable privacy vulneratbilities of such kind of GNNs, we reveal the threats that can potentially break the privacy of linear graph representations. Lastly, it remains to investigate that whether the inclusion of nonlinearity can mitigate the threat. While answering this question is hard in theory, we conducted extensive experiments demonstrating that nonlinear GNNs are often more vulnerable than linear GNNs.
>
> ## Q3: Rationale of studying SERA
> We chose SERA as the object of our analysis because of it is training-free, and requires a moderate level of adversary knowledge (node embeddings). There exists attacks with higher empirical attacking performance than SERA that utilizes additional knowledge (i.e., a certain part of input graph has already been compromised [1]) and a more sophisticated attacking procedure (i.e., training a shadow model [1]). In this paper, we take a first step toward establishing the vulnerabilities of graph representations in a theoretically principled way by studying the performance of SERA. For those attacks that are more sophisticated than SERA, it is an interesting question to study the improvement of attacking performance by quantitatively characterize the advantage of side information or auxiliary training, which is beyond the scope of our paper and we leave them for future explorations.
>
> ## Q4: Accuracy of experiment
> We agree with you that the definition of feature homophily is not suitable as a proper metric for describing the correlation between edge existence and node feature similarity. We include this metric primarily because this one is defined in previous works [2] and we list it mainly for completeness. The metric that indeed reflect the correlation is the $\widehat{A}^{\text{FS}}$ in table 1, which stands for tha AUC score between feature similarity and edge existence, that takes both edges and non-edges into account. We base most of our empirical findings regarding table 1 on the $\widehat{A}^{\text{FS}}$ metric. Please kindly refer to the experimental observations in section 7.1
>
> ## Q5: Explaination of the statement in line 286
> We appologize for making the statement too difficult to parse, and we shall restate the claim in revisions of our paper. By this statement, we want to express the following:
> - When $d$ is small (so that the assumptions in theorem 4.1 no longer holds), the claims in theorem 4.1 no longer holds empirically, as the attacking performances are limited (i.e., AUC score $\le 80\\%$).
> - We further explain why the phenomenon of limited attacking performance in small $d$ regime stems from: In our analysis, one primary mathematical tool is the concentration of inner products of two Gaussian vectors, which is highly dependent on the dimension of the two vectors (i.e., the feature dimension). When the concentration is insufficient (a consequence of small $d$), our analysis would then be no longer correct and this partly explains why the attacking performance is limited in small $d$ regimes.
>
> [1] He, Xinlei, et al. "Stealing links from graph neural networks." 30th USENIX security symposium (USENIX security 21). 2021.
> [2] Luan, Sitao, et al. "When do graph neural networks help with node classification? investigating the homophily principle on node distinguishability." Advances in Neural Information Processing Systems 36 (2023).

---

### Official Review · Reviewer_VCKp · 2024-07-10

**Soundness:** 2
**Presentation:** 2
**Contribution:** 2
**Rating:** 5
**Confidence:** 4

**Summary:**

The paper studies the performance of similarity-based edge reconstruction attacks (SERA) for graph representations, considering two particular similarity measures (cosine and correlation).

The main contributions are presented in Theorem 4.1 and Theorem 5.1, which analyze the performance of SERA on graph representations (without privacy-preservation techniques applied), and in Theorem 6.1, which considers noise aggregation (NAG) for privacy-preservation.

**Strengths:**

See my comments in Weaknesses

**Weaknesses:**

My concerns are as follows:

While I appreciate the efforts in deriving generalization bounds for this specific attack model, the implications of this paper remain insignificant to me. The results in Theorem 4.1 and Theorem 5.1 seem too straightforward, as they can be immediately inferred from the perspective of detection theory. For instance, focusing on the linear graph neural network in Equation (1), Theorem 4.1 essentially states that the accuracy of correlation detection grows with increasing samples. There are certainly existing works that have already characterized the statistical performance of link prediction in more general and profound ways, rather than fixing the detection method and linear aggregation as assumed in Theorems 4.1 and 5.1. Please refer to the papers "Revisiting Link Prediction: A Data Perspective" and "Statistical Guarantees for Link Prediction using Graph Neural Networks."

Theorem 5.1 also appears trivial. For example, the paper "GAP: Differentially Private Graph Neural Networks with Aggregation Perturbation" had already characterized the level of differential privacy for edge prediction. There is extensive literature connecting detection accuracy with differential privacy. For example, you can derive mutual information and then obtain the detection error bound by applying Fano's inequality (assuming the prior distribution of the graph representations, as stated in your paper). My question is, what is the new observation from Theorem 6.1?

**Questions:**

Lines 109-118: The description of the two-party attack model is confusing. I had to read the model description in the Appendix to understand it. Please keep the writing concise and precise to improve readability.

Lines 31-36: Can you explain how "feature similarity may serve as a confounding factor, potentially impacting the efficacy of similarity-based attacks"? This statement contradicts the previous sentence.

Line 156: $\Theta$ is not defined.

**Limitations:**

The authors adequately addressed the limitations in the Appendix.

---

> ### Author Rebuttal · Authors · 2024-08-05
>
> Thank you for your valuable comments. We appreciate your mentioning the two related works and we will include them in the related works of our paper with careful discusssions. As both of them focus on link prediction, we would like to make the following clarification first:
>
> ## In theory, link prediction is related to, but different from edge reconstruction
> Theoretically speaking, the problem of link prediction is formulated as the design of some algorithm $\mathcal{A}$ that takes input a known graph topology $A$ and some auxiliary features $X$ and output a predictor that is able to infer either missing edges in the training graph or generalizes to unseen graphs. Speaking in the context of statistical learning, the goal of link prediction is generalization. However, edge reconstruction is understood as an **inverse problem**[6] that is formulated as the design of some algorithm $\mathcal{A}$ that takes input a graph embedding with some knowledge on its generation process, and outputs an estimated topology that most possibly yields the graph embedding. The goal of edge reconstruction is not generalization as it recovers input graph in an instance-wise fashion (therefore, **we think you probably misunderstood the type of our analysis by refering to it as generalization bounds**). The difference here is somewhat reminiscent of that between the study of regression and compressive sensing: The techniques required in the analysis may have overlaps, but the two problems are different and require different types of analysis.
>
> Next we address your questions in detail:
> ## Q1: Are theorem 4.1 and 5.1 trivial, based on recent developments on link prediction?
> Now that we have explained the difference between link prediction and edge reconstruction, we state here why the theoretical developments in the two related works do not imply our establishments:
> - In [1], the authors focused on dataset properties that may influence the performance of link prediction. The analysis therein is conducted on certain types of random graphs, but the guarantees do not involve any specific algorithms or performance measures. Therefore we do not see any implications of [1] regarding our theoretical interest.
> - In [2], the authors derived generalization bounds of linear GNN under a **moderately sparse graphon model**, i.e., the edge density is of the order $\Omega(\log n / n)$. As we have previously mentioned, the generalization bounds of link prediciton is very different from edge recovery error bounds which is basically an inverse problem. Meanwhile, theorem 4.1 applies to graphs that are potentially much more sparse than those considered in [2] (please refer to assumption (i) in theorem 4.1. Besides, analyzing graphon models in sparse regimes as in our paper, i.e., $p_{uv} = o(\log n / n)$ is highly nontrivial [5]).  Therefore, both the analytical setup and goal are different between [2] and our paper.
>
>
> Regarding your comment that theorem 4.1 "essentially states that the accuracy of correlation detection grows with increasing samples", it is worth mentioning that in most of the theory developments in machine learning we need some sort of complexity that decreases with sample size and this should not be considered as essential implications of theory. Indeed, an important message of theorem 4.1 and 5.1 is that **sparsity plays a key role of edge recovery** and we are not aware of any previous works that noticed this factor.
>
> ## Q2: On theorem 6.1 and experiments
> Theorem 6.1 is mainly used as a statement that ascertains the protection guarantee of NAG against a wide range of adversaries, it slightly improves previous results [3] by allowing a more flexible choice of victim GNNs. ([3] only considers parameter-free GNN with summation pooling). Theorem 6.1 is indeed much easier to prove than theorem 4.1 and 5.1, and we do not consider theorem 6.1 to be our main theoretical contribution. However, we are interested in whether SERA can be utilized as a privacy auditing tool that empirically elicits the privacy level of NAG which are guaranteed by theorem 6.1------Such kind of empirical investigation is often considered as important research problems: For example, there has been considerable effort in designing MIA to audit DPSGD[4], and in such kind of study, we would need a theoretical privacy guarantee in the first place, which is what theorem 6.1 does.(We will include a more detailed discussion on privacy auditing in our revisions) According to our empirical findings, there are cases when SERA is ineffective but the theoretical privacy level according to theorem 6.1 is vacuous, thereby implying the limitations of SERA as a privacy auditing tool. In section 6, theorem 6.1 also serves as the motivation of our experiments.
>
> Hopefully we have addressed your theoretical concerns, and we would like to have an in-depth discussion with you. Please kindly let us know if you still have any questions regarding our theoretical contributions.
>
> [1] Mao, Haitao, et al. "Revisiting link prediction: A data perspective." arXiv preprint arXiv:2310.00793 (2023).
> [2] Chung, Alan, Amin Saberi, and Morgane Austern. "Statistical Guarantees for Link Prediction using Graph Neural Networks." arXiv preprint arXiv:2402.02692 (2024).
> [3] Sajadmanesh, Sina, et al. "{GAP}: Differentially Private Graph Neural Networks with Aggregation Perturbation." 32nd USENIX Security Symposium (USENIX Security 23). 2023.
> [4] Nasr, Milad, et al. "Adversary instantiation: Lower bounds for differentially private machine learning." 2021 IEEE Symposium on security and privacy (SP). IEEE, 2021.
> [5] Xu, Jiaming. "Rates of convergence of spectral methods for graphon estimation." International Conference on Machine Learning. PMLR, 2018.
> [6] Pasdeloup, Bastien, et al. "Graph reconstruction from the observation of diffused signals." 2015 53rd Annual Allerton Conference on Communication, Control, and Computing (Allerton). IEEE, 2015.

---

> > ### Author Response · Authors · 2024-08-13
> > **Please let us know if you have further concerns**
> >
> > Thank you for taking the time to review our paper. We genuinely appreciate your comments and believe they will help enhance our work. As the discussion period is drawing to a close, we would still like to engage with you further about our theoretical contributions. Please let us know if you have any additional concerns or questions.

---

> > > ### Comment · Reviewer_VCKp · 2024-08-14
> > >
> > > Thanks for your response.
> > >
> > > I'm quite confused by your explanation about the difference between link prediction and edge reconstruction. For me, they are just different names of topology inference.
> > >
> > > My concern is that the contribution of your results (which is built upon the classical detection theory (with random graph embeddings as inputs) does not seem significant to me.
> > >
> > > The sparsity assumption makes sure that the number of valid parameters of the adjacency matrix is limited so that the estimation accuracy can increase with increasing samples.
> > >
> > > I will raise the score but with reservation of the above concerns.

---

### Official Review · Reviewer_M6Kg · 2024-07-12

**Soundness:** 3
**Presentation:** 1
**Contribution:** 3
**Rating:** 6
**Confidence:** 3

**Summary:**

The paper studies to which degree graph representations are vulnerable to similarity-based edge reconstruction attacks (SERA).
SERAs encompass a variety of attacks used to recover the structure of a graph, where a SERA guesses that an edge exists between pairs of nodes that have similar embeddings.
The paper considers linear GNNs and highlights how SERAs are particularly successful in reconstructing large and sparse graphs.
On the other hand, the authors show that edge recovery is less successful when considering graphs generated from a stochastic block model.
Additionally, the paper presents an analysis and discussion of noisy aggregation as a mitigation technique against SERAs.

**Strengths:**

The paper presents an interesting overview of the strengths and limitations of edge reconstruction attacks based on similarity.
The motivation is clear and the paper is generally well structured.
The results on the theoretical performance of SERAs on both sparse and dense synthetic graphs are accompanied by an empirical evaluation.

**Weaknesses:**

**Assumptions**

The assumptions in Theorem 4.1 require further discussion.
While the authors mention in Remark 4.2 that the assumptions may not "consistently align with practical scenarios", it is unclear in which cases they may align at all with a practical scenario.
In section 7.1, you state that the linear model you analyse theoretically can be used as a proxy for a non-linear counterpart.
I therefore wonder: to which degree do the assumptions in Theorem 4.1 hold on real datasets?



**Empirical evaluation**

The readability of the he empirical results in Section 7 could be improved.
Specifically, the plots in Figure 1 and Figure 2 are very small, and the labels too tiny.
$\hat{A}^\text{SERA}$ in Table 1 is not defined, and so is the expression "trained" and "non-trained", in this context.
With respect to the results themselves, in Section 7.1 you claim that "SERA is able to achieve near-perfect reconstruction of all edges only in the 'large $d$, small $L$' regime".
Figure 1a, however, seems to show that for large $d$ the attack AUROC is high for all depths $L$.
These results would benefit from further comments and a better visualization of the results, as it is at the moment not easy to infer numerical values form the colour gradients in Figure 1.


**Language clarity**

The language is at times excessively verbose and difficult to parse.
For instance, in the description of the threat model in line 116, the "objectives ascribed to the adversary" are described as "decidedly audacious", as they aim at the "potential elucidation of the entire suite of edges" of an attacked graph.
These sentences are not only difficult to read, but also vague.
I would strongly recommend a more dry writing style which is better suited to convey technical content as, despite the good general organization of the paper, the current style hinders ease of read and is the main motivation for my low presentation score.



**Other comments**
* The brackets used for citations should, generally, not be interpreted as part of the sentence. So, e.g., in line 68, "proposed by [10]" -> "proposed by Duddu et al. [10]".
* Line 80, "don't" -> "do not".
* Line 152. "be a universal" -> "be universal".
* "related works" -> "related work".
* References to theorems, sections, etc., are inconsistently reported with both lower-case or upper-case initials.
* Several equations, particularly in the appendix, should be better typeset to help readability. Specifically, several parenthesis are not adjusted for the size of the expressions they contain.

**Questions:**

* In Remark 4.2 you state that the requirement of plolylogarithmic growth of the feature dimension is a byproduct of your proof. Can then this requirement be removed? How?
* In your theoretical investigation you do not use homophily to derive your results. While this is an interesting perspective as you bound the attack performance if homophily is not an assumption, what can you say when homophily _is_ an assumption?

**Limitations:**

The paper discusses limitations in the appendix.
It would preferable to expand the discussion of limitations in the main body of the paper.
The discussion of future work is completely deferred to the appendix, and would benefit as well from a dedicated paragraph in the main body of the paper.

---

> ### Author Rebuttal · Authors · 2024-08-05
>
> Thank you for your valuable comments and advices, we will integrate your suggestions into revised versions of our paper. Below we address some specific points:
>
> ## Q1: Practicality of assumptions in theorem 4.1
> According to our understanding, the primary concern is to what extent the assumption $d = \Omega(\text{polylog}(n))$ holds in practice. In our real-world experiments, the Cora and Citeseer datasets could be loosely regarded to satisty the assumption (Cora: $n=2708, d=1433$, Citeseer: $n=3327, d=3703$). Yet satisfiability of this polylog dimension relation seems not to be a necessary condition for SERA to succeed (at least empirically): For example, the Amazon Products and Reddit dataset are much larger in scale with much fewer features, yet the performance of SERA are fairly strong even for $L=5$.
> Regarding your question about whether the polylog factor can be removed, we think that the lower bound on $d$ can be improved to have a smaller exponent over $\log n$ (regarding the $6L+2$ in our paper) via a better handling of probabilistic arguments. But currently we do not know how to avoid the $(\log n)^{O(L)}$ dependence, which is an exciting future direction.
>
> ## Q2: Clarifications of empirical evaluations
> We will revise our graphical presentation in figure 1 in revised versions of the paper by adding visible score marks for a better illustration. Regarding your concern about the correctness of our claim in "Large $d$, small $L$" regimes, please kindly refer to our attached pdf file which shows the numerical values of SERA on Erdos Renyi graph with feature dimensions $d \in \\{512, 1024, 2048\\}$. The results demonstrates that the performance peaks at $L=3$ and reduces significantly for $L \ge 8$ across all the setups. It is true that the attack AUROC may still be as large as $95\\%$ when $n$ is large, but the performance drop is also statistically significant as opposed to near $100\\%$ AUROC for $L=3$. Therefore the empircal observations align with our theory findings.
> Regarding the missing defnitions in table 1, please kindly refer to our global response for an overview of the design of this table. Specifically, $\widehat{A}^{\text{SERA}}$ denotes the reconstruction is based on SERA over graph representations (we will change it to SERA for clarity in revisions), and the term **trained** and **non-trained** are used to indicate how the weight of the underlying victim GNN is obtained, either via random initialization or well-trained.
>
> ## Q3: Possiblity of incorporating homophily into our theory
> This is a very interesting question. In the setup of theorem 4.1 in our paper, we allow the edge generation probability $p_{uv}$ to depend on features $x_u$ and $x_v$ in an *arbitrary* fashion, which naturally includes the case of homophily. According to our understanding of your question, you might be suggesting the exploration of shaper reconstruction bounds that quantitatively incorporate a homophily measure into the bound that drastically change the complexity. This is a rather challenging task as a good quantitative characterization of homophily in this setup is non-trivial in the first place. While this question is beyond the scope of our paper, it warrants a careful study in the future.

---

> > ### Comment · Reviewer_M6Kg · 2024-08-12
> >
> > I thank you very much for your detailed answers and the additional experimental results. I think my current rating, considering the revision work necessary to improve readability, is appropriate for your submission and I will thus keep it as is.
> >
> > While the impact of a contribution has, of course, a degree of subjectivity, I nevertheless want to say that, differently to what reviewer VCKp points out, I find your analysis of the success/failure modes SERA and their link to sparsity to be valuable. I thus encourage further investigation in similar directions, to which I am looking forward!

---

> > > ### Author Response · Authors · 2024-08-12
> > > **Response by the authors**
> > >
> > > Thank you very much for the valuable responses. We really appreciate your insightful comments!

---

### Author Rebuttal · Authors · 2024-08-05

We thank the reviewers for their thought-provoking comments. We believe these valuable comments will lead to improvements of our paper. As we noticed some common concerns among the reviews, we provide some clarifications below:

## About our analysis on linear GNN
In appendix F.3 of our paper, we acknowledge that our analysis under linear GNN is a limitation of our work. The mathematical challenge in extending our analysis to nonlinear GNN stems from the relationships between node representations $h_v^{(L)}$'s and node features $x_v$'s. Under nonlinear GNNs, it becomes prohibitively hard to compute the non-trivial upper or lower bounds for the correlations between $h_u^{(L)}$ and $h_v^{(L)}$ for either $u \neq v$ or $u = v$.
Additionally, so far as we have noticed, many previous non-asymptotic theoretical developments on multi-layer GNNs are based on the linear GNN model [1, 2].
While theoretically analyzing nonlinear GNNs is challenging, we provide empirical evidences in section 7 (and more specifically in Appendix E.2) that systematically compare the performances of linear GNN and 4 prevailing nonlinear GNNs (GCN, SAGE, GAT and GIN) over 8 benchmark datasets with varying homophily level (See table 3 for a detailed report). The attack performances exhibit strong correlation between linear GNN and nonlinear GNNs across datasets. Besides, we observe from experimental results that *linear GNN often exhibits slightly weaker attack performance than nonlinear GNNs*. As we have shown that edge information is provably vulnerable even under linear GNN, we believe an analogue with nonlinear GNNs should hold and we left these developments to future explorations.

## Some clarifications of tables and figures
We thank the reviewers for pointing out illustration issues in figure 1 and table 1 of the paper. We provide some clarifications below:
1. [**Figure 1**] We present a detailed list of attack performance numbers in the attached pdf for $d \in \\{512, 1024, 2048\\}$ (corresponding to figure 1(a)). We will include a number-marked grid in revisions of the paper for improved illustrations.
2. [**Table 1**] The rationale behind the design of table 1 is that we want to verify the effectiveness as well as the robustness of SERA against varying *dataset characteristics* as well as *training dynamics*. To measure dataset characteristics (regarding homophily level), we use the feature homophily metric as well as the AUROC of feature similarity against edge existence, which we denote as $\widehat{A}^\text{FS}$. To systematically investigate the impact of traning dynamics, we conduct two set of experiments where the GNN weights are obtained either via random initializations (non-trained in table 1 and table 3), or via a standard training procedure (trained in table 1 and table 3) which we detail in appendix E.2 of the paper.

## About the writing style
We thank the reviewers for advices on our writing style and we will carefully polish our writing by enhancing clairty and conciseness.

[1] Wu, Xinyi, et al. "A Non-Asymptotic Analysis of Oversmoothing in Graph Neural Networks." The Eleventh International Conference on Learning Representations.
[2] Chung, Alan, Amin Saberi, and Morgane Austern. "Statistical Guarantees for Link Prediction using Graph Neural Networks." arXiv preprint arXiv:2402.02692 (2024).

---

### Decision · Program_Chairs · 2024-09-25

**Decision:**

Accept (poster)

**Comment:**

The paper discusses the ability to reconstruct a graph from representations learned by GRL techniques, theoretically and empirically.  The reviewers appreciated the analysis and results, but also highlighted the significant limitation that the theoretical results assume linearity (although empirical results do not).